# Overcoming the curse of dimensionality with Laplacian regularization in semi-supervised learning

**Vivien Cabannes**
ENS – INRIA – PSL
Paris, France
vivien.cabannes@gmail.com

**Loucas Pillaud-Vivien**
EPFL
Lausanne, Switzerland

**Francis Bach**
ENS – INRIA – PSL
Paris, France

**Alessandro Rudi**
ENS – INRIA – PSL
Paris, France

## Abstract

As annotations of data can be scarce in large-scale practical problems, leveraging unlabelled examples is one of the most important aspects of machine learning. This is the aim of semi-supervised learning. To benefit from the access to unlabelled data, it is natural to diffuse smoothly knowledge of labelled data to unlabelled one. This induces to the use of Laplacian regularization. Yet, current implementations of Laplacian regularization suffer from several drawbacks, notably the well-known curse of dimensionality. In this paper, we provide a statistical analysis to overcome those issues, and unveil a large body of spectral filtering methods that exhibit desirable behaviors. They are implemented through (reproducing) kernel methods, for which we provide realistic computational guidelines in order to make our method usable with large amounts of data.

In the last decade, machine learning has been able to tackle amazingly complex tasks, which was mainly allowed by computational power to train large learning models on large annotated datasets. For instance, ImageNet is made of tens of millions of images, which have all been manually annotated by humans [16]. The greediness in data annotation of such a current learning paradigm is a major limitation. In particular, when annotation of data demands in-depth expertise, relying on techniques that require zillions of labelled data is not viable. This motivates several research streams to overcome the need for annotations, such as self-supervised learning for images or natural language processing [17]. Aiming for generality, semi-supervised learning is the most classical one, assuming access to a vast amount of input data, but among which only a scarce percentage is labelled. To leverage the presence of unlabelled data, most semi-supervised techniques assume a form of low-density separation hypothesis, as detailed in the recent review of van Engelen and Hoos [52], and illustrated by state-of the-art models [6, 54]. This hypothesis assumes that the function to learn from the data varies smoothly in highly populated regions of the input space, but might vary more strongly in scarcely populated areas, or that the decision frontiers between classes lie in regions with low-density. In such a setting, it is natural to enforce constraints on the variations of the function to learn. While semi-supervised learning is an important learning framework, it has not provided as much exciting realizations as one could have expected. This might be related to the fact that it is classically approached through graph-based Laplacian, a technique that does not scale well with the dimension of the input space [5].

**Paper organization.** In Section 1, we motivate Laplacian regularization, and recall drawbacks of naive implementations. These limitations are overcome in Section 2 where we expose a theoretically principled path to derive well-behaved algorithms. More precisely, we unveil a vast class of estimates

35th Conference on Neural Information Processing Systems (NeurIPS 2021).

based on spectral filtering. We turn to implementation in Section 3 where we provide realistic guidelines to ensure scalability of the proposed algorithms. Statistical properties of our estimators are stated in Section 4.

**Contributions.** They are two folds. (*i*) Statistically, we explain that Laplacian regularization can be properly leveraged based on functional space considerations, and that those considerations can be turned into concrete implementations thanks to kernel methods. As a result, we provide consistent estimators that exhibit fast convergence rates under a low density separation hypothesis, and that, in particular, do not suffer from the curse of dimensionality. (*ii*) Computationally, we avoid dealing with large matrices of derivatives by providing a low-rank approximation that allows to deal with $n^\gamma \log(n) \times n^\gamma \log(n)$ matrices, with a parameter $\gamma \in (0, 1]$ depending on the regularity of the problem, instead of $n(d + 1) \times n(d + 1)$ matrices, thus cutting down to $\mathcal{O}(\log(n)^2 n^{1+2\gamma} d)$ the potential $\mathcal{O}(n^3 d^3)$ training cost.

**Related work.** Interplays between graph theory and machine learning were proven successful in the 2000s [47]. The seminal paper of Zhu et al. [62] introduced graph-Laplacian as a transductive method in the context of semi-supervised learning. A smoothing variant was proposed by [60], which is coherent with the fact that enforcing constraints on labelled points leads to spikes [1]. Interestingly, graph Laplacians do converge to diffusion operators linked with the weighted Laplace Beltrami operator [24, 21]. However, these local diffusion methods are known to suffer from the curse of dimensionality [5]. That is, local averaging methods are intuitive learning methods that have been used for more than half a century [20]. Yet, those methods do not scale well with the dimension of the input space [56]. This is related with the fact that to cover $[0, 1]^d$, we need $\varepsilon^{-d}$ balls of radius $\varepsilon$. Interestingly, if the function to learn is $m$ times differentiable with smooth partial derivatives, it is possible to leverage more information from function evaluations and overcome the curse of dimensionality when $m \gtrsim d$. This property is related to covering numbers (*a.k.a.* capacity) of Sobolev spaces [28] and is leveraged by (reproducing) kernel methods [49, 10]. The crux of this paper is apply this fact to Laplacian regularization techniques. Note that derivative with reproducing kernel methods in machine learning have already been considered in different settings by [61, 44, 19].

# 1 Laplacian regularization

In this section, we introduce the notations and concepts related to the semi-supervised learning regression problem, noting that most of our results extend to any convex loss beyond least-squares. We motivate and describe Laplacian regularization that will allow us to leverage the low-density separation hypothesis. We explain statistical drawbacks usually linked with Laplacian regularization, and discuss on how to circumvent them.

In the following, we denote by $\mathcal{X} = \mathbb{R}^d$ the input space, $\mathcal{Y} = \mathbb{R}$ the output space, and by $\rho \in \Delta_{\mathcal{X} \times \mathcal{Y}}$ the joint distribution on $\mathcal{X} \times \mathcal{Y}$. For simplicity, we assume that $\rho$ has compact support. In the following, we denote by $\rho_{\mathcal{X}}$ the marginal of $\rho$ over $\mathcal{X}$, and by $\rho|_x$ the conditional distribution of $Y$ given $X = x$. As usual, for $p \in \mathbb{N}^*$, $L^p(\mathbb{R}^d)$ is the space of functions $f$ such that $f^p$ is integrable. Moreover, we define usual Sobolev spaces: for $s \in \mathbb{N}$, $W^{s,p}(\mathbb{R}^d)$ stands for the space of functions whose weak derivatives of order $s$-th are in $L^p(\mathbb{R}^d)$. When $p = 2$, they have a Hilbertian structure and we denote, $H^s(\mathbb{R}^d) = W^{s,2}(\mathbb{R}^d)$ these Hilbertian spaces. Ideally, we would like to retrieve the mapping $g^* : \mathcal{X} \to \mathcal{Y}$ defined as

$$g^* = \underset{g \in L^2(\rho_{\mathcal{X}})}{\arg\min} \, \mathbb{E}_{(X,Y) \sim \rho} \left[ \|g(X) - Y\|^2 \right] = \underset{g \in L^2(\rho_{\mathcal{X}})}{\arg\min} \, \|g - g_\rho\|_{L^2(\rho_{\mathcal{X}})}^2 = g_\rho, \tag{1}$$

where $g_\rho : \mathcal{X} \to \mathcal{Y}$ is defined as $g_\rho(x) = \mathbb{E}[Y \,|\, X = x]$. In semi-supervised learning, we assume that we do not have access to $\rho$ but we have access to $n$ independent samples $(X_i)_{i \le n} \sim \rho_{\mathcal{X}}^{\otimes n}$, among which we have $n_\ell$ labels $Y_i \sim \rho|_{X_i}$ for $i \le n_\ell$, with $n_\ell$ potentially much smaller than $n$. In other terms, we have $n_\ell$ supervised pairs $(X_i, Y_i)_{i \le n_\ell}$, and $n - n_\ell$ unsupervised samples $(X_i)_{n_\ell < i \le n}$. While we restrict ourselves to real-valued regression for simplicity, our exposition indeed applies generically to partially supervised learning. In particular, it can be used off-the-shelve to complement the approaches of [7, 8] as we detailed in Appendix A.

| Semi-supervision setting | Baseline: $\lambda = 0$, $\mu = 1$ | Result: $\lambda = 1$, $\mu = 1/n$ |
|:---:|:---:|:---:|
| 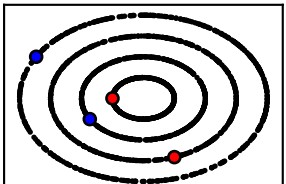 | 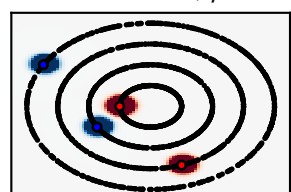 | 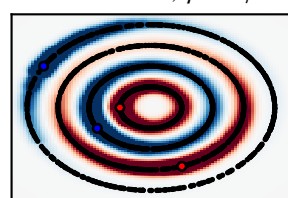 |

**Figure 1:** Motivating example. (Left) We suppose given $n = 2000$ points in $\mathcal{X} = \mathbb{R}^2$, represented as black dots, spanning 4 concentric circles. Among those points are $n_\ell = 4$ labelled points, with labels being either 1 represented in red, and $-1$ represented in blue. In this setting, it is natural to assume that $g^*$ should be constant on each circles, which can be encoded as $\|\nabla g^*\| = 0$ on supp $\rho_{\mathcal{X}}$. (Middle) Kernel ridge regression estimate based on the labelled points with Gaussian kernel of bandwidth $\sigma = .2r$, $r$ being the radius of the innermost circle. (Right) Laplacian regularization reconstruction. The reconstruction is based on approximate empirical risk minimization with $p = n$, which ensures a computational complexity of $O(p^2 n d)$, instead of $O(n^3 d^3)$ needed to recover the exact empirical risk minimizer (5).

## 1.1 Diffusion operator $\mathcal{L}$

In order to leverage unlabelled data, we will assume that $g^*$ varies smoothly on highly populated regions of $\mathcal{X}$, and might vary highly on low density regions. For example, this is the case when data are clustered in well separated regions of space, and labels are constant on clusters. This is captured by the fact that the Dirichlet energy

$$\int_{\mathcal{X}} \|\nabla g^*(x)\|^2 \rho_{\mathcal{X}}(\mathrm{d}x) = \mathbb{E}_{X \sim \rho_{\mathcal{X}}} \left[ \|\nabla g^*(X)\|^2 \right] =: \left\| \mathcal{L}^{1/2} g \right\|^2_{L^2(\rho_{\mathcal{X}})}, \tag{2}$$

is assumed to be small. Because the quadratic functional (2) will play a crucial role in our exposition, we define $\mathcal{L}$ as the self-adjoint operator on $L^2(\rho_{\mathcal{X}})$, extending the operator on $H^1(\rho_{\mathcal{X}})$ representing this functional. Under mild assumptions on $\rho_{\mathcal{X}}$, $\mathcal{L}^{-1}$ can be shown to be a compact operator, which we will assume in the following. In essence, we will assume that if we have a lot of unlabelled data and $\left\| \mathcal{L}^{1/2} g \right\|$ can be well approximated for any function $g$, then we do not need a lot of labelled data to estimate correctly $g^*$. To illustrate this, at one extreme, if we know that $\left\| \mathcal{L}^{1/2} g^* \right\| = 0$, then $g^*$ is known to be constant on each connected component of $\rho_{\mathcal{X}}$ so that, along with the knowledge of $\rho_{\mathcal{X}}$, only a few labelled points would be sufficient to recover perfectly $g^*$. We illustrate those considerations on Figure 1.

## 1.2 Drawbacks of naive Laplacian regularization

Following the motivations presented precedently, it is natural to consider the regularized objective and solution defined, for $\lambda > 0$, as

$$\begin{aligned} g_\lambda &= \underset{g \in H^1(\rho_{\mathcal{X}})}{\arg\min} \ \mathbb{E}_{(X,Y) \sim \rho} \left[ \|g(X) - Y\|^2 \right] + \lambda \, \mathbb{E}_{X \sim \rho_{\mathcal{X}}} \left[ \|\nabla g(X)\|^2_{\mathbb{R}^d} \right] \\ &= \underset{g \in H^1(\rho_{\mathcal{X}})}{\arg\min} \ \|g - g_\rho\|^2_{L^2(\rho_{\mathcal{X}})} + \lambda \left\| \mathcal{L}^{1/2} g \right\|^2_{L^2(\rho_{\mathcal{X}})} = (I + \lambda \mathcal{L})^{-1} g_\rho. \end{aligned} \tag{3}$$

This regularization has nice properties. In particular, for small $\lambda$, it can be seen as a first order approximation of the heat equation solution $e^{-\lambda \mathcal{L}} g_\rho$, which represents the temperature profile at time $t = \lambda$, instantiated with the initial profile $g_\rho$, and with $\rho_{\mathcal{X}}$ modelling the thermal conductivity. It also has interpretations in term of random walk and Langevin diffusion [40, 48]. In a word, $g_\lambda$ is the diffusion of $g_\rho$ with respect to the density $\rho_{\mathcal{X}}$, which relates to the idea of diffusing labelled data with respect to the intrinsic geometry of the data, which is the idea captured by [62].

However, from a learning perspective, Eq. (3) is linked with the prior that $g^*$ belongs to $H^1(\rho_{\mathcal{X}})$, a prior that is not strong enough to overcome the curse of dimensionality as we saw in the related work section. Moreover, assuming we have enough unsupervised data to suppose known $\rho_{\mathcal{X}}$, and therefore $\mathcal{L}$, Eq. (3) leads to the naive empirical estimate $g_{(\text{naive})} \in \arg\min_{g:\mathcal{X} \to \mathbb{R}} \sum_{i=1}^{n_\ell} \|g(X_i) - Y_i\|^2 + n_\ell \lambda \left\| \mathcal{L}^{1/2} g \right\|^2$. While the definition of $g_{(\text{naive})}$ could seem like a great idea, in fact, such an estimate $g_{(\text{naive})}$ is known to be mostly constant and spiking to interpolate the data $(X_i, Y_i)$ as soon as $d > 2$

[37]. This is to be related with the capacity of the space associated with the pseudo-norm $\left\|\mathcal{L}^{1/2}g\right\|$ in $L^2$. This capacity, related to $H^1$, is too large for the Laplacian regularization term to constraint $g_{\text{(naive)}}$ in a meaningful way. In other terms, we need to regularize with stronger penalties.

## 1.3 Stronger regularization

In this subsection, we discuss techniques to overcome the issues encountered with $g_{\text{(naive)}}$. Those techniques are based on functional space constraints or on spectral filtering techniques.

**Functional spaces.** A solution to overcome the capacity issue of $H^1$ in $L^2$ is to constrain the estimate of $g^*$ to belong to a smaller functional space. In the realm of graph Laplacian, [1] proposed to solve this problem by considering the $r$-Laplacian regularization reading $\Omega_r = \int_{\mathcal{X}} \|\nabla g(X)\|^r \rho(\mathrm{d}x)$, with $r > d$. In essence, this restricts $g$ to live in $W^{1,r}(\rho_{\mathcal{X}})$ for $r > d$, and allows to avoid spikes associated with $g_{\text{(naive)}}$. However considering high power of the gradient is likely to introduce instability (think that $d$ is the potentially really big dimension of the input space), and from a learning perspective, the capacity of $W^{1,r}$, which compares to the one of $H^2$, is still too big.

In this paper, we will rather keep the diffusion operator $\mathcal{L}$, and add a second penalty to reduce the space in which we look for the solution. With $\mathcal{G}$ an Hilbert space of functions, we could look for, with $\mu > 0$ a second regularization parameter

$$g_{\lambda,\mu} = \operatorname*{arg\,min}_{g:\mathcal{G}\cap H^1(\rho_{\mathcal{X}})} \|g - g_\rho\|^2_{L^2(\rho_{\mathcal{X}})} + \lambda \left\|\mathcal{L}^{1/2}g\right\|^2_{L^2(\rho_{\mathcal{X}})} + \lambda\mu \|g\|^2_{\mathcal{G}}. \qquad (4)$$

This formulation restricts $g_{\lambda,\mu}$ to belong both to $H^1(\rho_{\mathcal{X}})$ (thanks to the term in $\lambda$) and $\mathcal{G}$ (thanks to the term in $\mu$). In particular the resulting space $H^1(\rho_{\mathcal{X}}) \cap \mathcal{G}$ to which $g_{\lambda,\mu}$ belongs, has a smaller capacity in $L^2$ than the one of $\mathcal{G}$ in $L^2$. In practice, we do not have access to $\rho$ and $\rho_{\mathcal{X}}$ but to $(X_i, Y_i)_{i \leq n_\ell}$ and $(X_i)_{i \leq n}$, and we might consider the empirical estimator defined through empirical risk minimization

$$g_{n_\ell,n} = \operatorname*{arg\,min}_{g\in\mathcal{G}} n_\ell^{-1} \sum_{i=1}^{n_\ell} \|g(X_i) - Y_i\|^2 + \lambda n^{-1} \sum_{i=1}^{n} \|\nabla g(X_i)\|^2 + \lambda\mu \|g\|^2_{\mathcal{G}}. \qquad (5)$$

For example, we could consider $\mathcal{G}$ to be the Sobolev space $H^m(\mathrm{d}x)$. Note the difference between $\mathcal{G}$ linked with $\mathrm{d}x$, the Lebesgue measure, that is known, and $\mathcal{L}$ linked with $\rho_{\mathcal{X}}$, the marginal of $\rho$ over $\mathcal{X}$, that is not known. In this setting, the regularization $\|\mathcal{L}^{1/2}g\|^2 + \mu\|g\|^2_{\mathcal{G}}$ reads $\int_{\mathcal{X}} \|Dg(x)\|^2 \rho_{\mathcal{X}}(\mathrm{d}x) + \mu \int_{\mathcal{X}} \sum_{\alpha=0}^{m} \|D^\alpha g(x)\|^2 \mathrm{d}x$. Because of the size of $H^m$ in $L^2$, this allows for efficient approximation of $g_{\lambda,\mu}$ based on empirical risk minimization. In particular, if $n = +\infty$, we expect the minimizer (5) to converge toward $g_{\lambda,\mu}$ at rates in $L^2$ scaling similarly to $n_\ell^{-m/d}$ in $n_\ell$. To complete the picture, depending on a prior on $g_\rho$, $g_{\lambda,\mu}$ might exhibit good convergence properties towards $g_\rho$ as $\lambda$ and $\mu$ go to zero. This contrasts with the problem encountered with $g_{\text{(naive)}}$. Those considerations are exactly what reproducing kernel Hilbert space will provide, additionally with a computationally friendly framework to perform the estimation. Note that quantity similar to $g_{\lambda,\mu}$ were considered in [61, 44].

**Spectral filtering.** Without looking for higher power-norm, [37] proposed to overcome the capacity issue by considering approximation of the operator $\mathcal{L}$ based on the graph-based technique provided by [4, 14] and to reduce the search of $g_{n_\ell}$ on the space spanned by the first few eigenvectors of the Laplacian. In particular, on Figure 1, $g^*$ could be searched in the null space of $\mathcal{L}$, that is, among functions that are constant on each connected component of $\operatorname{supp} \rho_{\mathcal{X}}$. This technique exhibits two parts, the "unsupervised" estimation of $\mathcal{L}$ that will depend on the total number of data $n$, and the "supervised" search for $g_\rho$ on the first few eigenvectors of $\mathcal{L}$ that will depend on the number of labels $n_\ell$. While, at first sight, this technique seems to be completely different than Tikhonov regularization (4), it can be cast, along with gradient descent, into the same *spectral filtering* framework [30]. This point of view enables the use of a wide range of techniques offered by spectral manipulations on the diffusion operator $\mathcal{L}$.

This paper is motivated by the fact that current well-grounded semi-supervised learning techniques are implemented based on graph-based Laplacian, which is a local averaging method that does not leverage smartly functional capacity. In particular, as recalled earlier, graph-based Laplacian is known

to suffer from the curse of dimensionality, in the sense that the convergence of the empirical estimator $\widehat{\mathcal{L}}$ towards the $\mathcal{L}$ exhibits a rate of convergence of order $\mathcal{O}(n^{-1/d})$ with $d$ the dimension of the input space $\mathcal{X}$ [24]. In this work, we will bypass this curse of dimensionality by looking for $g$ in a smooth universal reproducing kernel Hilbert space, which will lead to efficient empirical estimates.

## 2   Spectral Filtering with Kernel Laplacian

In this section, we approach Laplacian regularization from a functional analysis perspective. We first introduce kernel methods and derivatives in reproducing kernel Hilbert space (RKHS). We then translate the considerations provided in Section 1.3 in the realm of kernel methods.

### 2.1   Kernel methods and derivatives evaluation maps

In this subsection, we introduce kernel methods (see [3, 46, 49] for more details Consider $(\mathcal{H}, \langle \cdot, \cdot \rangle_{\mathcal{H}})$ a reproducting kernel Hilbert space, that is a Hilbert space of functions from $\mathcal{X}$ to $\mathbb{R}$ such that the evaluation functionals $L_x : \mathcal{H} \to \mathbb{R}; g \to g(x)$ are continuous linear forms for any $x \in \mathcal{X}$. Such forms can be represented by $k_x \in \mathcal{H}$ such that, for any $g \in \mathcal{H}$, $L_x(g) = \langle k_x, g \rangle_{\mathcal{H}}$. A reproducing kernel Hilbert space can alternatively be defined from a symmetric positive semi-definite kernel $k : \mathcal{X} \to \mathcal{X} \to \mathbb{R}$, that is a function such that for any $n \in \mathbb{N}$ and $(x_i)_{i \leq n} \in \mathcal{X}^n$ the matrix $(k(x_i, x_j))_{i,j}$ is symmetric positive semi-definite, by building $(k_x)_{x \in \mathcal{X}}$ such that $k(x, x') = \langle k_x, k_{x'} \rangle_{\mathcal{H}}$. From a learning perspective, it is useful to use the evaluation maps to rewrite $\mathcal{H} = \{ g_\theta : x \to \langle k_x, \theta \rangle_{\mathcal{H}} \,|\, \theta \in \mathcal{H} \}$. As such, kernel methods can be seen as "linear models" with features $k_x$, allowing to parameterize large spaces of functions [34]. In the following, we will differentiate $\theta$ seen as an element of $\mathcal{H}$ and $g_\theta$ seen as its embedding in $L^2$. To make this distinction formal, we define the embedding $S : (\mathcal{H}, \langle \cdot, \cdot \rangle_{\mathcal{H}}) \hookrightarrow (L^2(\rho_{\mathcal{X}}), \langle \cdot, \cdot \rangle_{L2}); \theta \to g_\theta$, as well as its adjoint $S^\star : L^2(\rho_{\mathcal{X}}) \to \mathcal{H}$.

Given a linear parametric model of functions $g_\theta(x) = \langle \theta, k_x \rangle_{\mathcal{H}}$, it is possible to compute derivatives of $g_\theta$ based on derivatives of the feature vector – think of $\mathcal{H} = \mathbb{R}^p$ and of $k_x = \varphi(x)$ as a feature vector with $\varphi : \mathbb{R}^d \to \mathbb{R}^p$. For $\alpha \in \mathbb{N}^d$, with $|\alpha| = \sum_{i \leq d} \alpha_i$, we have the following equality of partial derivatives, when $k$ is $2|\alpha|$ times differentiable,

$$D^\alpha g_\theta(x) = \langle \theta, D^\alpha k_x \rangle, \qquad \text{where} \qquad D^\alpha = \frac{\partial^{|\alpha|}}{(\partial x_1)^{\alpha_1} (\partial x_2)^{\alpha_2} \cdots (\partial x_d)^{\alpha_d}}.$$

Here and $D^\alpha k_x$ has to be understood as the partial derivative of the mapping of $x \in \mathcal{X}$ to $k_x \in \mathcal{H}$, which can be shown to belong to $\mathcal{H}$ [61]. In the following, we assume that $k$ is twice differentiable with continuous derivatives, and will make an extensive use of derivatives of the form $\partial_i k_x = \partial k_x / \partial x_i$ for $i \leq d$ and $x \in \mathcal{X}$. Note that, as well as we can describe completely the Hilbertian geometry of the space $\text{Span} \{ k_x \,|\, x \in \mathcal{X} \}$ through $k(x, x') = \langle k_x, k'_x \rangle$, for $x, x' \in \mathcal{X}$, we can describe the Hilbertian geometry of $\text{Span} \{ k_x \,|\, x \in \mathcal{X} \} + \text{Span} \{ \partial_i k_x \,|\, x \in \mathcal{X} \}$, through

$$\partial_{1,i} k(x, x') = \langle \partial_i k_x, k_{x'} \rangle_{\mathcal{H}}, \qquad \text{and} \qquad \partial_{1,i} \partial_{2,j} k(x, x') = \langle \partial_i k_x, \partial_j k_{x'} \rangle_{\mathcal{H}},$$

where $\partial_{1,i}$ denotes the partial derivative with respect to the $i$-th coordinates of the first variable. This echoes to so-called "representer theorems".

**Example 1** (Gaussian kernel). *A classical kernel is the Gaussian kernel, also known as radial basis function, defined for $\sigma > 0$ as the following $k$, and satisfying, for $i \neq j$, the following equalities,*

$$k(x, x') = \exp\left( -\frac{\|x - x'\|^2}{2\sigma^2} \right), \qquad \partial_{1,i} \partial_{2,j} k(x, y) = -\frac{(x_i - y_i)(x_j - y_j)}{\sigma^4} k(x, y),$$

$$\partial_{1,i} k(x, y) = -\frac{(x_i - y_i)}{\sigma^2} k(x, y), \qquad \partial_{1,i} \partial_{2,i} k(x, y) = \left( \frac{1}{\sigma^2} - \frac{(x_i - y_i)^2}{\sigma^4} \right) k(x, y),$$

*where $x_i$ designs the $i$-th coordinates of the vector $x \in \mathcal{X} = \mathbb{R}^d$.*

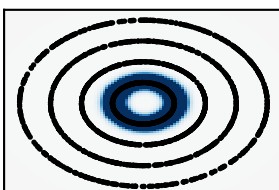
Eigen vector #1 ($e_1$)

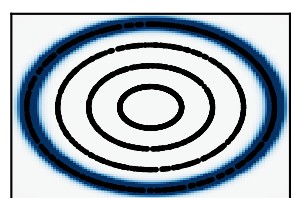
Eigen vector #4 ($e_4$)

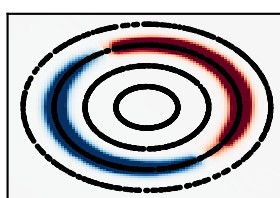
Eigen vector #8 ($e_8$)

**Figure 2:** Few of the first generalized eigenvectors of $(\hat{\Sigma}; \hat{L} + \mu I)$ (with $\mu = 1/n$). The first four eigenvectors correspond to constant functions on each circle, as shown with $e_1$ and $e_4$. The few eigenvectors after correspond to second harmonics localized on a single circle as shown with $e_8$.

## 2.2 Tikhonov, spectral filtering and dimensionality reduction

Given the kernel $k$, its associated RKHS $\mathcal{H}$ and $S$ the embedding of $\mathcal{H}$ in $L^2$, we rewrite Eq. (4) under its "parameterized" version

$$g_{\lambda,\mu} = S \arg\min_{\theta \in \mathcal{H}} \left\{ \|S\theta - g_\rho\|^2_{L^2(\rho_{\mathcal{X}})} + \lambda \left\| \mathcal{L}^{1/2} S\theta \right\|^2_{L^2(\rho_{\mathcal{X}})} + \lambda\mu \|\theta\|^2_{\mathcal{H}} \right\}. \tag{4}$$

Do not hesitate to refer to Table 1 to keep track of notations. In the following, we will use that $\left\| \mathcal{L}^{1/2} S\theta \right\|^2_{L^2(\rho_{\mathcal{X}})} + \mu \|\theta\|^2_{\mathcal{H}} = \left\| (S^\star \mathcal{L} S + \mu I)^{1/2} \theta \right\|^2_{\mathcal{H}}$. This equality explains why we consider $\mu\lambda$ instead of $\mu$ in the last term. In the RKHS setting, the study of Eq. (4) unveils the three operators $\Sigma, L,$ and $I$ on $\mathcal{H}$, (indeed $g_{\lambda,\mu} = S \arg\min_{\theta \in \mathcal{H}} \{\theta^\star (\Sigma + \lambda L + \lambda\mu)\theta - 2\theta^\star S^\star g_\rho\}$) where $I$ is the identity, and, as we detail in Appendix C,

$$\Sigma = S^\star S = \mathbb{E}_{X \sim \rho_{\mathcal{X}}} [k_X \otimes k_X], \qquad \text{and} \qquad L = S^\star \mathcal{L} S = \mathbb{E}_{X \sim \rho_{\mathcal{X}}} \left[ \sum_{i=1}^d \partial_j k_X \otimes \partial_j k_X \right]. \tag{6}$$

Regularization and spectral filtering have been well-studied in the inverse-problem literature. In particular, the regularization Eq. (4) is known to be linked with the generalized singular value decomposition of $[\Sigma; L + \mu I]$ (see, *e.g.*, [18]), which is linked to the generalized eigenvalue decomposition of $(\Sigma, L + \mu I)$ [22]. We derive the following characterization of Eq. (4), whose proof is reported in Appendix D.

**Proposition 1.** *Let $(\lambda_{i,\mu})_{i \in \mathbb{N}} \in \mathbb{R}^{\mathbb{N}}, (\theta_{i,\mu})_{i \in \mathbb{N}} \in \mathcal{H}^{\mathbb{N}}$ be the generalized eigenvalue decomposition of the pair $(\Sigma, L + \mu I)$, that is $(\theta_{i,\mu})$ generating $\mathcal{H}$ and such that for any $i, j \in \mathbb{N}$, $\Sigma\theta_{i,\mu} = \lambda_{i,\mu}(L + \mu I)\theta_{i,\mu}$, and $\langle \theta_{i,\mu}, (L + \mu I)\theta_{j,\mu} \rangle = \mathbf{1}_{i=j}$. Eq. (4) can be rewritten as*

$$g_{\lambda,\mu} = \left( \sum_{i \in \mathbb{N}} \psi(\lambda_{i,\mu}) S\theta_{i,\mu} \otimes S\theta_{i,\mu} \right) g_\rho = \sum_{i \in \mathbb{N}} \psi(\lambda_{i,\mu}) \langle S^\star g_\rho, \theta_{i,\mu} \rangle S\theta_{i,\mu}, \tag{7}$$

*with $\psi : \mathbb{R}_+ \to \mathbb{R}; x \to (x + \lambda)^{-1}$. Eq. (7) should be seen as a specific instance of spectral filtering based on a filter function $\psi : \mathbb{R}_+ \to \mathbb{R}$.*

Interestingly, the generalized eigenvalue decomposition of the pair $(\Sigma, L + \mu I)$ was already considered by Pillaud-Vivien [40] to estimate the first eigenvalue of the Laplacian. Moreover, Pillaud-Vivien [41] suggests to leverage this decomposition for dimensionality reduction based on the first eigenvectors of the Laplacian. As well as Eq. (4) contrasts with graph-based semi-supervised learning techniques, this dimensionality reduction technique contrasts with methods based on graph Laplacian provided by [4, 14]. Remarkably, the semi-supervised learning algorithm that consists in using the unsupervised data to perform dimensionality reduction based on the Laplacian eigenvalue decomposition, before solving a small linear regression problem on the small resulting space, can be seen as a specific instance of spectral filtering, based on regularization by thresholding/cutting-off eigenvalue, which corresponds to $\psi : x \to x^{-1}\mathbf{1}_{x > \lambda}$ for a given threshold $\lambda > 0$ in Eq. (7).

## 3 Implementation

In this section, we discuss on how to practically implement estimates for Eq. (7) based on empirical data $(X_i, Y_i)_{i \leq n_\ell}$ and $(X_i)_{n_\ell < i \leq n}$. We first review how we can approximate the integral operators of

Eq. (6) based on data. We then discuss on how to implement our methods practically on a computer. We end this section by considering approximations that allow to cut down high computational costs associated with kernel methods involving derivatives.

---

**Algorithm 1:** Empirical estimates based on spectral filtering.

---

**Data:** $(X_i, Y_i)_{i \leq n_\ell}$, $(X_i)_{n_\ell < i \leq n}$, a kernel $k$, a filter $\psi$, a regularizer $\mu$
**Result:** $\hat{g}_p$ through $c \in \mathbb{R}^p$ defining $\hat{g}_p(x) = \sum_{i=1}^{n} c_i k(x, X_i) = k_x^\star T_a c$
Compute $S_n T_a = (k(X_i, X_j))_{i \leq n, j \leq p} \in \mathbb{R}^{n \times p}$ in $\mathcal{O}(pn)$
Compute $Z_n T_a = (\partial_{1,j} k(X_l, X_i))_{(j \leq d, l \leq n), i \leq p} \in \mathbb{R}^{nd \times p}$ in $\mathcal{O}(pnd)$
Build $T_a^\star \hat{\Sigma} T_a = n^{-1}(S_n T_a)^\top (S_n T_a)$ in $\mathcal{O}(p^2 n)$
Build $T_a^\star \hat{L} T_a = n^{-1}(Z_n T_a)^\top (Z_n T_a)$ in $\mathcal{O}(p^2 nd)^a$
Build $T_a^\star T_a = (k(X_i, X_j))_{i,j \leq p} \in \mathbb{R}^{p \times p}$ in $\mathcal{O}(1)$ as a partial copy of $S_n T_a$
Get $(\lambda_{i,\mu}, u_{i,\mu})_{i \leq n}$ the generalized eigenelements of $(T_a^\star \hat{\Sigma} T_a, T_a^\star(\hat{L} + \mu I) T_a)$ in $\mathcal{O}(p^3)$
Get $b = T_a^\star \hat{\theta} = (n_\ell^{-1} \sum_{i=1}^{n_\ell} Y_i k(X_i, X_j))_{j \leq p} \in \mathbb{R}^p$ in $\mathcal{O}(pn_\ell)$
Return $c = \sum_{i=1}^{n} \psi(\lambda_i) u_i u_i^\top b \in \mathbb{R}^p$ in $\mathcal{O}(p^3)$.

---

$^a$Building this matrix can be avoided by using the generalized singular value decomposition rather than the generalized eigenvector decomposition. Implemented with Lapack, such a procedure will also requires $O(p^2 nd)$ floating point operations, but with a smaller constant in the big $O$ [22].

## 3.1 Integral operators approximation

The classical empirical risk minimization in Eq. (5) can be understood as the plugging of the approximate distributions $\hat{\rho} = n_\ell^{-1} \sum_{i=1}^{n_\ell} \delta_{X_i} \otimes \delta_{Y_i}$ and $\hat{\rho}_\mathcal{X} = n^{-1} \sum_{i=1}^{n} \delta_{X_i}$ instead of $\rho$ and $\rho_\mathcal{X}$ in Eq. (4). It can also be understood as the same replacement when dealing with integral operators, leading to the three following important quantities to rewrite Eq. (7),

$$\hat{\Sigma} := n^{-1} \sum_{i=1}^{n} k_{X_i} \otimes k_{X_i}, \quad \hat{L} := n^{-1} \sum_{i=1}^{n} \sum_{j=1}^{d} \partial_j k_{X_i} \otimes \partial_j k_{X_i}, \quad \hat{\theta} := \widehat{S^\star g_\rho} := n_\ell^{-1} \sum_{i=1}^{n_\ell} Y_i k_{X_i}. \quad (8)$$

It should be noted that while considering $n$ in the definition of $\hat{\Sigma}$ is natural from the spectral filtering perspective, to make it formally equivalent with the empirical risk minimization (5), it should be replaced by $n_\ell$. Eq. (8) allows to rewrite Eq. (7) without relying on the knowledge of $\rho$, by considering $(\hat{\lambda}_{i,\mu}, \hat{\theta}_{i,\mu})$ the generalized eigenvalue decomposition of $(\hat{\Sigma}, \hat{L})$ and considering

$$\hat{g} = \sum_{i \in \mathbb{N}} \psi(\hat{\lambda}_{i,\mu}) \left\langle \widehat{S^\star g_\rho}, \hat{\theta}_{i,\mu} \right\rangle S \hat{\theta}_{i,\mu}, \quad (9)$$

We present the first eigenvectors (after plunging them in $L^2$ through $S$) of the generalized eigenvalue decomposition of $(\hat{\Sigma}, \hat{L} + \mu I)$ on Figure 2. The first eigenvectors allow to recover the null space of $\mathcal{L}$. This explains clearly the behavior on the right of Figure 1.

## 3.2 Matrix representation and approximation of operators

Currently, we are dealing with operators $(\hat{\Sigma}, \hat{L})$ and vectors (*e.g.*, $\hat{\theta}$) in the Hilbert space $\mathcal{H}$. It is natural to wonder on how to represent this on a computer. The answer is the object of representer theorems (see Theorem 1 of [61]), and consists in noticing that all the objects introduced are actually defined in, or operate on, $\mathcal{H}_n + \mathcal{H}_{n,\partial} \subset \mathcal{H}$, with $\mathcal{H}_n = \text{Span}\{k_{X_i} \mid i \leq n\}$ and $\mathcal{H}_{n,\partial} = \text{Span}\{\partial_j k_{X_i} \mid i \leq n, j \leq d\}$. This subspace of $\mathcal{H}$ is of dimension at most $n(d+1)$ and if $T : \mathbb{R}^p \to \mathcal{H}_n + \mathcal{H}_{n,\partial}$ (with $p \leq n(d+1)$) parameterizes $\mathcal{H}_n + \mathcal{H}_{n,\partial}$, our problem can be cast in $\mathbb{R}^p$ by considering the $p \times p$ matrices $T^\star \hat{\Sigma} T$ and $T^\star(\hat{L} + \mu I) T$ instead of the operators $\hat{\Sigma}$ and $\hat{L} + \mu I$. The canonical representation consists in taking $p = n(d+1)$ and considering for $c \in \mathbb{R}^{n(d+1)}$, the mapping $T_c c = \sum_{i=1}^{n} c_{i0} k_{X_i} + \sum_{j=1}^{d} c_{ij} \partial_j k_{X_i}$ [61, 44].

This exact implementation implies dealing and finding the generalized eigen value decomposition of $p \times p$ matrices with $p = n(d+1)$, which leads to computational costs in $\mathcal{O}(n^3 d^3)$, which can be prohibitive. Two solutions are known to cut down prohibitive computational costs of kernel methods.

Both methods consist in looking for a space that can be parameterized by $\mathbb{R}^p$ for a small $p$ and that approximates well the space $\mathcal{H}_n + \mathcal{H}_{n,\partial} \subset \mathcal{H}$. The first solution is provided by random features [42]. It consists in approximating $\mathcal{H}$ with a space of small dimension $p \in \mathbb{N}$, linked with an explicit representation $\varphi : \mathcal{X} \to \mathbb{R}^p$ that approximate $k(x, x') \simeq k_\varphi(x, x') = \langle \varphi(x), \varphi(x') \rangle_{\mathbb{R}^p}$. In theory, it substitutes the kernel $k$ by $k_\varphi$. In practice, all computations can be done with the explicit feature $\varphi$.

**Approximate solution.** The second solution, which we are going to use in this work, consists in approximating $\mathcal{H}_n + \mathcal{H}_{n,\partial}$ by $\mathcal{H}_p = \operatorname{Span}\{k_{X_i}\}_{i \leq p}$ for $p \leq n$. This method echoes the celebrated Nyström method [55], as well as the Rayleigh–Ritz method for Sturm–Liouville problems. In essence, [45] shows that, when considering subsampling based on leverage score, $p = n^\gamma \log(n)$, with $\gamma \in (0, 1]$ linked to the "size" of the RKHS and the regularity of the solution, is a good enough approximation, in the sense that it only downgrades the sample complexity by a constant factor. In theory, we know that the space $\mathcal{H}_p$ will converge to $\mathcal{H} = \operatorname{Closure}\operatorname{Span}\{k_x\}_{x \in \operatorname{supp}\rho_{\mathcal{X}}}$ as $p$ goes to infinity. In practice, it means considering the approximation mapping $T_a : \mathbb{R}^p \to \mathcal{H}; c \to \sum_{i=1}^p c_i k_{X_i}$, and dealing with the $p \times p$ matrices $T_a^\star \Sigma T_a$ and $T_a^\star L T_a$. It should be noted that the computation of $T_a^\star L T_a$ requires to multiply a $p \times nd$ matrix by its transpose. Overall, training this method can be done with $\mathcal{O}(p^2 nd)$ basic operations, and inference with this method can be done in $\mathcal{O}(p)$. The saving cost of this approximate method is huge: without compromising the precision of our estimator, we went from $O(n^3 d^3)$ run time complexities to $O(\log(n)^2 n^{1+2\gamma} d)$ computations, with $\gamma$ possibly very small. Similarly, the memory cost went from $O(n^2 d^2)$ down to $O(nd + n^{2\gamma})$.[1]

# 4 Statistical analysis

In this section, we are interested in quantifying the risk of the learnt mapping $\hat{g}$. We study it through the generalization bound, which consists in obtaining a bound on the averaged excess risk $\mathbb{E}_{\text{data}} \|\hat{g} - g_\rho\|_{L^2}^2$. In particular, we want to answer the following points.

1. How, and under which assumptions, Laplacian regularization boost learning?
2. How the excess of risk relates to the number of labelled and unlabelled data?

In terms of priors, we want to leverage a low-density separation hypothesis. In particular, we can suppose that when diffusing $g_\rho$ with $e^{-t\mathcal{L}}$ we stay close to $g_\rho$, or that $g_\rho$ is supported on a finite dimensional space of functions on which $\|\mathcal{L}^{1/2} g\|$ (which measures the variation of $g$) is small. Both those assumptions can be made formal by assuming the $g_\rho$ is supported by the first eigenvectors of the diffusion operator $\mathcal{L}$.

**Assumption 1** (Source condition). *$g_\rho$ is supported on a finite dimensional space that is left stable by the diffusion operator $\mathcal{L}$. In other terms, if $(e_i) \in (L^2)^{\mathbb{N}}$ are the eigenvectors of $\mathcal{L}$, there exists $r \in \mathbb{N}$, such that $g_\rho \in \operatorname{Span}\{e_i\}_{i \leq r}$.*

We will also assume that the diffusion operator $\mathcal{L}$ can be well approximated by the RKHS associated with $k$. In practice, under mild assumptions, *c.f.* Appendix C, the eigenvectors of the Laplacian are known to be regular, in particular to belong to $H^m$ for $m \in \mathbb{N}$ bigger than $d$. As such, many classical kernels would allow to verify the following assumption.

**Assumption 2** (Approximation condition). *The eigenvectors $(e_i)$ of $\mathcal{L}$ belongs to the RKHS $\mathcal{H}$.*

We add one technical assumptions regarding the eigenvalue decay of the operator $\Sigma$ compared to the operator $L$, with $\preceq$ denoting the Löwner order (*i.e.*, for $A$ and $B$ symmetric, $A \preceq B$ if $B - A$ is positive semi-definite).

**Assumption 3** (Eigenvalue decay). *There exists $a \in [0, 1]$ and $c > 0$ such that $L \preceq c\Sigma^\alpha$.*

Note that, in our setting, $L$ is compact and bounded and Assumption 3 is always satisfied with $a = 0$. For translation-invariant kernel, such as Gaussian or Laplace kernels, based on considerations linking eigenvalue decay of operators with functional space capacities [49], under mild assumptions, we can take $a > 1 - 2/d$. We discuss all assumptions in more details in Appendix C.

To study the consistency of our algorithms, we can reuse the extensive literature on kernel ridge regression [10, 30]. This literature body provides an extensive picture on convergence rates relying

---

[1] Our code is available online at `https://github.com/VivienCabannes/partial_labelling`.

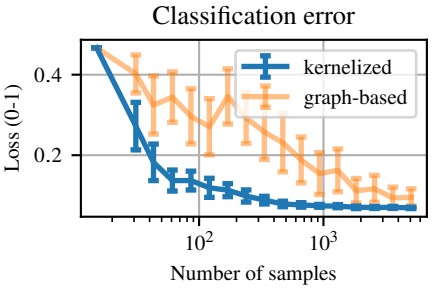
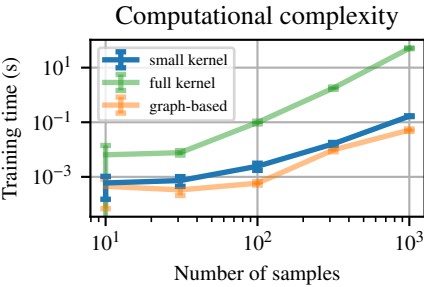

**Figure 3:** (Left) Comparison between our kernelized Laplacian method (Tikhonov regularization version with $\lambda = 1$, $\mu_n = n^{-1}$, $p = 50$) and graph-based Laplacian based on the same Gaussian kernel with bandwidth $\sigma_n = n^{-\frac{1}{d+4}} \log(n)$ as suggested by graph-based theoretical results [24]. We report classification error as a function of the number of samples $n$. The error is averaged over 50 trials, with errorbars representing standard deviations. We fixed the ratio $n_\ell/n$ to one tenth, and generated the data according to two Gaussians in dimension $d = 10$ with unit variance and whose centers are at distance $\delta = 3$ of each other (similar to the setting of [11, 29]). Our method discovers the structure of the data much faster than graph-based Laplacian (to get a 20% error we need 40 points, while graph-based need 700). (Right) Time to perform training with graph-based Laplacian in orange, with Algorithm 1 in blue (with the specification of the left figure), and with the naive representation in $\mathbb{R}^{n(d+1)}$ of the empirical minimizer Eq. (5) in green. When dealing with 1000 points, our algorithm, as well as graph-based Laplacian, can be computed in about one tenth of a second on a 2 GHz processor, while the naive kernel implementation requires 10 seconds. We show in Appendix B that this cut in costs is not associated with a loss in performance.

on various filters and assumptions of capacity, *a.k.a* effective dimension, and source conditions. Our setting is slightly different and showcases two specificities: (*i*) the eigenelements $(\lambda_{i,\mu}, \theta_{i,\mu})$ are dependent of $\mu$; (*ii*) the low-rank approximation in Algorithm 1 is specific to settings with derivatives. We end our exposition with the following convergence result, proven in Appendix E. Note that the dependency of $p$ in $n$ can be improved based on subsampling techniques that leverage expressiveness of the different $(k_{X_i})$ [45]. Moreover, universal consistency results could also be provided when the RKHS is dense in $H^1$, as well as convergence rates for other filters and laxer assumptions which we discuss in Appendix E (in particular, the source condition can be relaxed by considering the biggest $q \in (0, 1]$ such that $g \in \text{im } \mathcal{L}^q$).

**Theorem 1** (Convergence rates). *Under Assumptions 1, 2 and 3, for $n_\ell, n \in \mathbb{N}$, when considering the spectral filtering Algorithm 1 with $\psi_\lambda : x \to (x + \lambda)^{-1}$, there exists a constant $C$ independent of $n$, $n_\ell$, $\lambda$, $\mu$ and $p$ such that the estimate $\hat{g}_p$ defined in Algorithm 1 verifies*

$$\mathbb{E}_{\mathcal{D}_n}\left[\|\hat{g}_p - g_\rho\|_{L^2}^2\right] \leq C\left(\lambda^2 + \lambda\mu + \frac{\sigma_\ell^2 n_\ell^{-1} + n_\ell^{-2} + n^{-1}}{\lambda\mu} + \frac{\log(p)}{p} + \lambda\frac{\log(p)^a}{p^a}\right), \quad (10)$$

*with $\sigma_\ell^2$ is a variance parameter that relates to the variance of the variable $Y(I + \lambda\mathcal{L})^{-1}\delta_X$, inheriting its randomness from $(X, Y) \sim \rho$. In particular, when the ratio $r = n_\ell/n$ is fixed, with the regularization scheme $\lambda_n = \lambda_0 n^{-1/4}$, $\mu_n = \mu_0 n^{-1/4}$, for any $\lambda_0 > 0$ and $\mu_0 > 0$, and the subsampling scheme $p_n = p_0 n^s \log(n)$ for any $p_0 > 0$ and with $s = \max(1/2, 1/4a)$, there exists a constant $C'$ independent of $n$ and $n_\ell$ such that the excess of risk verifies*

$$\mathbb{E}_{\mathcal{D}_n}\left[\|\hat{g}_p - g_\rho\|_{L^2}^2\right] \leq C'(n^{-1/2} + \sigma_\ell^2 n_\ell^{-1/2}). \quad (11)$$

Theorem 1 answers the two questions asked at the beginning of this section. In particular, it characterizes the dependency of the need for labelled data to a variance parameter linked with the diffusion of observations $(X_i, Y_i)$ based on the density $\rho_{\mathcal{X}}$ through the operator $\mathcal{L}$. Finally, Theorem 1 is remarkable in that it exhibits no dependency to the dimension of $\mathcal{X}$ in the power of $n$ and $n_\ell$. This contrasts with graph-based Laplacian methods that do not scale well with the input space dimensionality [5, 24]. Indeed, Figure 3 shows the superiority of our method over graph-based Laplacian in dimension $d = 10$, with a mixture of Gaussians. We provide details as well as additional experiments in Appendix B.

## 5 Conclusion

Diffusing information or enforcing regularity through penalties on derivatives are natural ideas to tackle many machine learning problems. Those ideas can be captured informally with graph-based techniques and finite element differences, or captured more formally with the diffusion operator we introduced in this work. This formalization allowed us to shed lights on Laplacian regularization techniques based on statistical learning considerations. In order to make our method usable in practice, we provided strong computational guidelines to cut down prohibitive cost associated with a naive implementation of our methods. In particular, we were able to develop computationally efficient semi-supervised techniques that do not suffer from the curse of dimensionality.

This work paves the way to many extensions beyond semi-supervised learning. For example, in Appendix A, we describe its usefulness to the partial supervised learning problem, where minimizing the Dirichlet energy provide a learning principle, in order to bypass the restrictive non-ambiguity assumption usually made in this setup [15, 31, 7, 8]. Moreover, in the context of active learning, retaking the strategy of Karzand and Nowak [27], this energy provides a computationally-effective, theoretically-grounded, data-dependent score to select the next point to query. As such, follow-ups would be of interest to see how this introductory theoretical paper makes its way into the world of concrete applications.

### Acknowledgments and Disclosure of Funding

This work was funded in part by the French government under management of Agence Nationale de la Recherche as part of the "Investissements d'avenir" program, reference ANR-19-P3IA-0001 (PRAIRIE 3IA Institute). We also acknowledge support of the European Research Council (grants SEQUOIA 724063 and REAL 94790).

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
