**Ethical considerations**

This work aims at advancing our understanding of weakly supervised learning. Weakly supervised learning enrolls in the quest of an automated artificial intelligence, free from the need of human supervision. Automation, which is at the basis of computer science [51], is known to increase productivity at a reduced human labor cost, and is associated with several political/societal issues. In term of concrete applications, reducing the need for annotations is especially useful when humans reproduce biases when annotating data, or when the lack of output annotation restricts the outreach of a method (*e.g.*, learning to translate languages by collecting input/output pairs based on books already translated by humans can hardly be applied to languages with few written resources).

# A    Extensions: Least-square surrogate and partially supervised learning

In this section, we first show how our work can be extended to generic semi-supervised learning problem, beyond real-valued regression. This first extension is based on the least-square surrogate introduced by Ciliberto et al. [13] for structured prediction problems. We later show how our work can be extended to generic partially-supervised learning. This second extension is based on the work of Cabannes et al. [7].

## A.1    Structured prediction and least-square surrogate

Until now, we have considered the least-square problem with $Y \in \mathbb{R}$. Indeed, our work can be extended easily to a wide class of learning problem. Consider $\mathcal{Y}$ an output space, $\ell : \mathcal{Y} \times \mathcal{Y} \to \mathbb{R}$ a loss function, and keep $\mathcal{X} \subset \mathbb{R}^d$ and $\rho \in \Delta_{\mathcal{X} \times \mathcal{Y}}$. Suppose that we want to retrieve

$$f^* = \arg\min_{f:\mathcal{X} \to \mathcal{Y}} \mathcal{R}(f), \qquad \text{with} \qquad \mathcal{R}(f) = \mathbb{E}_{(X,Y)\sim\rho} \left[ \ell(f(X), Y) \right]. \tag{12}$$

Ciliberto et al. [13] showed that as soon as $\ell$ can be decomposed through two mappings $\varphi : \mathcal{Y} \to \mathcal{H}_{\mathcal{Y}}$ and $\psi : \mathcal{Y} \to \mathcal{H}_{\mathcal{Y}}$ with $\mathcal{H}_{\mathcal{Y}}$ a Hilbert space as $\ell(y, z) = \langle \varphi(y), \psi(z) \rangle_{\mathcal{H}_{\mathcal{Y}}}$, it is possible to leverage the least-square regression by considering the surrogate problem

$$g^* \in \arg\min_{g:\mathcal{X} \to \mathcal{H}_{\mathcal{Y}}} \mathbb{E}_{(X,Y)\sim\rho} \left[ \|g(X) - \varphi(Y)\|^2_{\mathcal{H}_{\mathcal{Y}}} \right]. \tag{13}$$

This surrogate problem relates to the original one through the decoding $d$ that relates a surrogate estimate $g : \mathcal{X} \to \mathcal{H}_{\mathcal{Y}}$ to an estimate of the original problem $f : \mathcal{X} \to \mathcal{Y}$ as $f = d(g)$ defined through, for $x \in \operatorname{supp} \rho_{\mathcal{X}}$,

$$f(x) = \arg\min_{z \in \mathcal{Y}} \langle \psi(z), g(x) \rangle_{\mathcal{H}_{\mathcal{Y}}}. \tag{14}$$

In the real-valued regression case, presented precedently, our estimates for $g_n$ can all be written as $g_n(x) = \sum_{i=1}^{n_\ell} \beta_i(x) Y_i$, where $\beta_i(x)$ is a function of the $(X_i)_{i \leq n}$, involving the kernel $k$ and its derivatives. Those estimates can be cast to vector-valued regression by considering coordinates-wise regression[2], which leads to $g_n(x) = \sum_{i=1}^{n_\ell} \beta_i(x) \varphi(Y_i)$, and to the original estimates, for any $x \in \operatorname{supp} \rho_{\mathcal{X}}$,

$$f_n(x) \in \arg\min_{z \in \mathcal{Z}} \sum_{i=1}^{n_\ell} \beta_i(x) \ell(z, Y_i). \tag{15}$$

The behavior of $f_n$ being independent of the decomposition $(\varphi, \psi)$ of $\ell$ was referred to as the loss trick. In particular, Ciliberto et al. [13] showed that convergence rates derived between $\|g_n - g^*\|_{L^2}$ does not change if we consider $g : \mathcal{X} \to \mathbb{R}$ or $g : \mathcal{X} \to \mathcal{H}_{\mathcal{Y}}$ and that those rates can be cast directly as convergence rates between $\mathcal{R}(f_n)$ and $\mathcal{R}(f^*)$ with $f_n = d(g_n)$ defined by Eq.(14). Moreover, when $\mathcal{Y}$ is a discrete output space, it is possible to get much better generalization bound on $\mathcal{R}(f_n) - \mathcal{R}^*$ by introducing geometrical considerations regarding $g^*$ and decision frontier between classes [9].

---

[2]To parameterize functions $g$ from $\mathcal{X}$ to $\mathcal{H}_{\mathcal{Y}}$, we can parameterized independently each coordinates $\langle g, e_i \rangle_{\mathcal{H}_{\mathcal{Y}}}$, for $(e_i)$ a basis of $\mathcal{H}_{\mathcal{Y}}$, by the space $\mathcal{G}$ – note that it is possible to generalize real-valued kernel to parameterize coordinates in a joint fashion [10]. The coordinate-wise parameterization corresponds to the tensorization $\mathcal{H}' = \mathcal{H}_{\mathcal{Y}} \otimes \mathcal{H}$ and to the parametric space $\mathcal{G}' = \{x \to \Theta k_x \,|\, \Theta \in \mathcal{H}'\}$ of functions from $\mathcal{X}$ to $\mathcal{H}_{\mathcal{Y}}$. $\mathcal{G}'$ naturally inherits of the Hilbertian structure of $\mathcal{H}'$, itself inherited from the structure of $\mathcal{H}$ and $\mathcal{H}_{\mathcal{Y}}$.

**Example 2** (Binary classification)**.** *This framework aims at generalizing well known surrogate considerations in the case of the binary classification. Binary classification corresponds to $\mathcal{Y} = \{-1, 1\}$, $\ell$ the $0 - 1$ loss. In this setting, $\mathcal{H}_\mathcal{Y} = \mathbb{R}$, $\varphi : \mathcal{Y} \to \mathbb{R}; y \to y$, and $\psi = -\varphi$. This definition verifies $\ell(y, z) = .5 - .5\varphi(y)^\top \varphi(z) \simeq \varphi(y)^\top \psi(z)$. This corresponds to the usual least-square surrogate, which is $\mathcal{R}_S(g) = \mathbb{E}[\|g(X) - Y\|^2]$, $g(x) = \mathbb{E}[Y \mid X = x]$ and $f = \text{sign } g$.*

**Beyond least-squares.** Considering a least-square surrogate assumes that retrieving $g^*$ (13) is the way to solve the original problem (12) and that the low-density separation hypothesis can be expressed as Assumption 1 being verified by $g^*$. We would like to point out that the low-density separation could be expressed under a much weaker form, which is that there exists $g$ such that $f^* = d(g)$ (14) and $g$ verifies Assumption 1. In particular, the cluster assumption [43] could be understood as assuming that $g = \varphi(f^*)$, the trivial embedding of $f^*$ in $\mathcal{H}_\mathcal{Y}$, is constant on clusters, with means that $g$ belongs to the kernel of the Laplacian operator $\mathcal{L}$. Yet, $g^* : x \to \mathbb{E}[\varphi(Y)|X = x]$, which depends on the labelling noise, could be really non-smooth, even under the cluster assumption. Those considerations are related to an open problem in machine learning, which is that we do not know what is the best statistical way (and the best surrogate problem) to solve the fully supervised binary classification problem [see *e.g.* 59]. However, many points introduced in the work could be retaken with other surrogate, could it be SVM (which leads to $g^* = \varphi(f^*)$, with $g^*$ minimizing the Hinge loss), softmax regression (used in deep learning) or others.

### A.2 Partially supervised learning

Partial supervision is a popular instance of weak supervision, which generalizes semi-supervised learning. It has been known under the name of partial labelling [15], superset learning [31], as well as learning with partial label [23], with partial annotation [32], with candidate labeling set [33] or with multiple label [26]. It encompasses many problems such as "classification with partial labels" [38, 15], "multilabelling with missing labels" [57], "ranking with partial ordering" [25], "regression with censored data" [50], "segmentation with pixel annotation" [53, 39], as well as instances of "action retrieval", especially on instructional videos [2, 35].

It consists, for a given input $x$, in not observing its label $y \in \mathcal{Y}$, but observing a set of potential labels $s \in 2^\mathcal{Y}$ that contains the labels ($y \in s$). Typically, if $\mathcal{Y}$ is the space $\mathfrak{S}_m$ of orderings between $m$ items (*e.g.* movies on a streaming website), for a given input $x$ (*e.g.* some feature vectors characterizing a user) $s$ might be specified by a partial ordering that the true label $y$ should satisfy (*e.g.* the user prefers romantic movies over action movies).

In this setting, it is natural to create consensus between the different sets giving information on $(y|x)$, which has been formalized mathematically by the infimum loss $(z, s) \in \mathcal{Y} \times 2^\mathcal{Y} \to \inf_{y \in s} \ell(z, y) \in \mathbb{R}$ for $\ell : \mathcal{Y} \times \mathcal{Y} \to \mathbb{R}$ a specified loss on the underlying fully supervised learning problem. This leads, for $\tau \in \Delta_{\mathcal{X} \times 2^\mathcal{Y}}$ encoding the distribution generating samples $(X, S)$, to the formulation $f^* \in \mathcal{F} = \arg\min_{f:\mathcal{X} \to \mathcal{Y}} \mathbb{E}_{(X,S) \sim \tau} [\inf_{Y \in S} \ell(f(X), Y)]$. To study this problem, a non-ambiguity assumption is usually made [15, 33, 31, 7, 8]. This is a very strong assumption to ensure that $\mathcal{F}$ is, in essence, a singleton. Highly adequate to this setting, the Laplacian regularization allows to relax this assumption, assuming that $\mathcal{F}$ can be big, but that we can discriminate between function in $\mathcal{F}$ by looking for the smoothest one in the sense defined by the Laplacian penalty. Moreover, the loss trick (15) allows to endow, in a off-the-shelf fashion, the recent work of Cabannes et al. [7, 8] on the partial supervised learning problem with our considerations on Laplacian regularization.

## B Experiments

### B.1 Low-rank approximation

Cutting computation cost thanks to low-rank approximation, as we did by going from the naive exact empirical risk minimizer $\hat{g}$ Eq. (5) to the smart implementation $\hat{g}_p$ Algorithm 1, is associated with a trade-off between computational versus statistical performance. This trade-off can be studied theoretically thanks to Theorem 1, which shows that under mild assumptions, considering $p = n^{1/2} \log(n)$ does not lead to any loss in performance, in the sense that the convergence rates in $n$, the number of samples, are only changed by a constant factor. We show on Figure 4 that in the setting of Figure 3, our low-rank approximation is not associated with a loss in performance. Actually low-rank

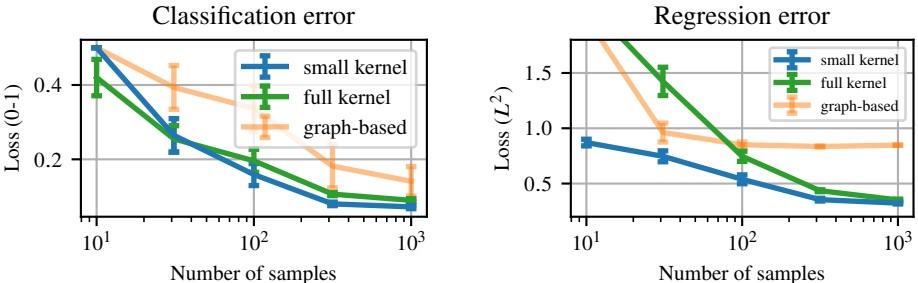

**Figure 4:** Cut in computation cost are not associated with a loss in performance. The estimate $\hat{g}_p$ Algorithm 1 (in blue), based on low-rank approximation that cut computation cuts performs as well as the exact computation of $\hat{g}$ Eq. (5). (Left) Classification error in the setting of Figure 3. (Right) Regression error in the same setting. The fact that the error of the graph-based method stalls around one, is due to the amplitude of the estimate being very small, which is coherent with behaviors described in [37].

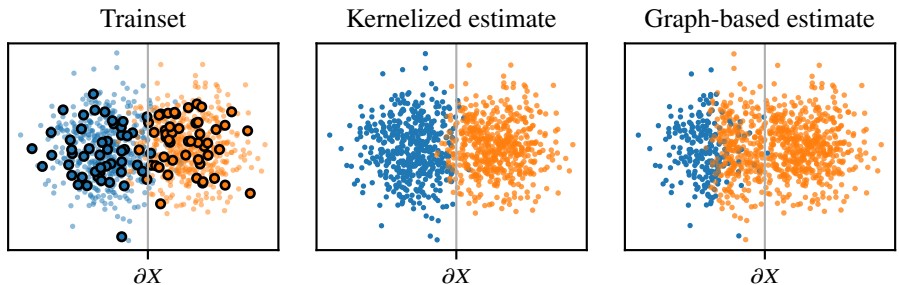

**Figure 5:** Setting of Figure 3 with $n = 1000$. (Left) Training set. We represent a cut of $\mathcal{X} \subset \mathbb{R}^d$ according to the two first coordinates $\{(x_1, x_2) \,|\, (x_1, x_2, \cdots, x_d) \in \mathcal{X}\}$. We have two Gaussians distribution with unit variance, one centered at $x = (0, 0, \cdots 0)$ and the other one centered at $x = (3, 0, \cdots, 0)$. One of the Gaussian distribution is associated to the blue class, the other one with the orange class. We consider $n = 1000$ unlabelled points, represented by small points, colored according to their classes, and $n_\ell = 100$ labelled points, represented in colour with black edges. (Middle) Reconstruction with our kernelized Laplacian methods. Our method uncovers correctly the structure of the problem, allowing to make a quite optimal reconstruction. The optimal decision frontier being illustrated by the grey line $\partial X$. (Right) Reconstruction with graph-Laplacian. The graph-Laplacian diffuses information too far away from what it should, leading to many incorrect guesses.

approximation can even be beneficial as it tends to lower the risk for overfitting as discussed by Rudi et al. [45].

## B.2 Comparison with graph-based Laplacian

One the main goal of this paper is to make people drop graph-based Laplacian methods and adopt our "kernelized" technique. As such, we would like to discuss in more detail our comparison with graph-based Laplacian. In particular, we will discuss how and why we choose the hyperparameters and the setting of Figure 3.

The setting of Figure 3 is the one of Figure 5, we considered two Gaussians with unit variance and whose centers are at distance $\delta = 3$ of each other. We chose Gaussians distributions as it is a well-understood setting. We chose $\delta = 3$ so that there is an mild overlap between the two distributions. For the bandwidth parameter, we considered $\sigma_n = Cn^{-\frac{1}{d+4}} \log(n)$ as this is known to be the optimal bandwidth for graph Laplacian [24]. We chose $C = 1$ as this leads to $\sigma_n$ of the order of $\delta$. We chose $\lambda = 1$ to enforce Laplacian regularization and $\mu_n = 1/n$, as this is a classical regularization parameter in RKHS. We did not cross-validate parameters in order to be fair with graph-Laplacian that do not have as much parameters as our kernel method. We compute the error in a transductive setting, retaking the exact problem and algorithm of Zhu et al. [62]. We choose $d = 10$, as we know

$$S(\hat{\Sigma}+\varepsilon)^{-1}\widehat{S^\star g_\rho}$$ 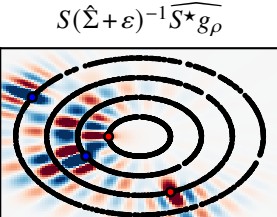 $$S(\hat{L}+\varepsilon)^{-1}\widehat{S^\star g_\rho}$$ 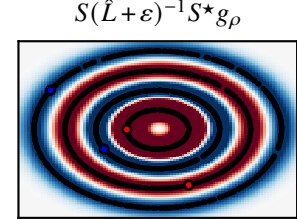

**Figure 6:** Usefulness of Laplacian regularization. We illustrate the reconstruction based on our spectral filtering techniques based on the sole use of the covariance matrix $\Sigma$ on the left, and on the sole use of the Laplacian matrix $L$ on the right. We see that the covariance matrix does not capture the geometry of the problem, which contrasts with the use of Laplacian regularization.

that this is a good dimension parameter in order to illustrate the curse of dimensionality phenomenon without needing too much data.[3]

### B.3 Usefulness of Laplacian regularization

It is natural to ask about the relevance of Laplacian regularization. To give convergence results, we have used Assumptions 1 and 2, which imply that $g^*$ belongs to the RKHS $\mathcal{H}$, and we got convergence rates in $n_l^{1/2}$, which is not better than the rates we could get with pure kernel ridge regression. In particular, our algorithm can be split between an unsupervised part that learn the penalty $\left\|\mathcal{L}^{1/2}g\right\|_{L^2(\rho_\mathcal{X})}^2$ and a supervised part, that solve the problem of estimating $g_\lambda$ from few labels $(X_i, Y_i)$ given the penalty associated to $\mathcal{L}$. But the same method can be used for pure kernel ridge regression: unsupervised data could be leveraged to learn the covariance matrix $\Sigma$ (6), and supervised data could be used to get $\widehat{S^\star g_\rho}$ to converge towards $S^\star g_\rho$. The same analysis would yield the same type of convergence rates. Yet the parameter $\sigma_\ell$ appearing in Theorem 1 would not be linked with the variance of $Y(I + \lambda\mathcal{L} + \lambda\mu K^{-1})^{-1}\delta_X$ but with the variance of $Y(I + \mu K^{-1})^{-1}\delta_X$. This is a key fact, the geometry of the covariance operator $\Sigma$ is not supposed to be that relevant to the problem, while the one of $L$ is. We illustrate this fact on Figure 6.

## C Central Operators

The paper makes an intensive use of operators. This section aims at providing details and intuitions on those operators, in order to help the reader. In particular, we discuss on Assumptions 1 and 2 and we prove the equality in Eq. (6).

### C.1 The diffusion operator

In this subsection, we discuss on the diffusion operator, and recall its basic properties.

The diffusion operator is a well-known operator in the realm of partial differential equation. Let us assume that $\rho_\mathcal{X}$ admit a smooth density $\rho_\mathcal{X}(dx) = p(x)dx$, say $p \in \mathcal{C}^2(\mathbb{R}^d)$, that cancels outside a domain $\Omega \subset \mathbb{R}^d$. Then the diffusion operator $\mathcal{L}$ can be explicitly written, for $g$ twice differentiable, as

$$\mathcal{L}g(x) = -\Delta g(x) + \frac{1}{p(x)}\left\langle \nabla p(x), \nabla g(x) \right\rangle.$$

---

[3]Note that our consistency result Theorem 1 describes a convergence regime that applies to a vast class of problems. Such a regime usually takes place after a certain number of data (depending on the value of the constant $C$). Before entering this regime, describing the error of our algorithm would require more precise analysis specific to each problem instance, eventually involving tools from random matrix theory.

**Table 1:** Notations

| Symbol | Description |
| --- | --- |
| $(X_i)_{i \leq n}$ | $n$ samples of input data |
| $(Y_i)_{i \leq n_l}$ | $n_l$ labels |
| $\rho$ | Distribution of $(X, Y)$ |
| $g_\rho$ | Function to learn (1) |
| $\lambda, \mu$ | Regularization parameters |
| $g_\lambda, g_{\lambda,\mu}$ | Biased estimates (3, 4) |
| $\hat{g}$ | Empirical estimate (5) |
| $\hat{g}_p$ | Empirical estimate with low-rank approximation (Algo. 1) |
| $\mathcal{H}$ | Reproducing kernel Hilbert space |
| $k$ | Reproducing kernel |
| $S$ | Embedding of $\mathcal{H}$ in $L^2$ |
| $S^\star$ | Adjoint of $S$, operating from $L^2$ to $\mathcal{H}$ |
| $\Sigma = S^\star S$ | Covariance operator on $\mathcal{H}$ |
| $K = SS^\star$ | Equivalent of $\Sigma$ on $L^2$ |
| $\mathcal{L}$ | Diffusion operator (*a.k.a* Laplacian) |
| $L = S^\star \mathcal{L} S$ | Restriction of the diffusion operator to $\mathcal{H}$ |
| $g$ | Generic element in $L^2$ |
| $\theta$ | Generic element in $\mathcal{H}$ |
| $\lambda_i$ | Generic eigen value |
| $e_i$ | Generic eigen vector in $L^2$ |

This follows from the fact that for $f$ once and $g$ twice differentiable, using Stokes theorem,

$$\langle f, \mathcal{L}g \rangle_{L^2(\rho_{\mathcal{X}})} = \langle \nabla f, \nabla g \rangle_{L^2(\rho_{\mathcal{X}})} = \langle \nabla f, p \nabla g \rangle_{L^2(\mathrm{d}x)}$$

$$= \int_{\mathcal{X}} \mathrm{div}(fp\nabla g) \, \mathrm{d}x - \langle f, \mathrm{div}(p\nabla g) \rangle_{L^2(\mathrm{d}x)} = - \langle f, \mathrm{div}(p\nabla g) \rangle_{L^2(\mathrm{d}x)}$$

$$= - \langle f, p^{-1} \mathrm{div}(p\nabla g) \rangle_{L^2(\rho_{\mathcal{X}})} = - \langle f, \mathrm{div}\, \nabla g + p^{-1}(\nabla p).\nabla g \rangle_{L^2(\rho_{\mathcal{X}})}.$$

Note that when the distribution is uniform on $\Omega$, the diffusion operator is exactly the opposite usual Laplacian operator $\Delta$. As for the Laplacian case, it can be shown that under mild assumption on $p$, whose smoothness properties directly translates to the smoothness properties of the boundary of $\Omega$, that the diffusion operator $\mathcal{L}$ has a compact resolvent (that is, for $\lambda \notin \mathrm{spec}(\mathcal{L})$, $(\mathcal{L} + \lambda I)^{-1}$ is compact). This is a standard result implied by a standard version of the famous Rellich-Kondrachov compactness embedding theorem: $H^2(\Omega)$ is compactly injected in $L^2(\Omega)$ whenever $\Omega$ is a bounded open with $C^2$-boundaries.

In such a setting, we can consider the eigenvalue decomposition of $\mathcal{L}^{-1}$, that is, $(\lambda_i, e_i) \in (\mathbb{R}_+ \times L^2)^{\mathbb{N}}$, with $(e_i)_{i \in \mathbb{N}}$ an orthonormal basis of $L^2$ and $(e_i)_{i \leq \dim \ker \mathcal{L}}$ generating the null space of $\mathcal{L}$, with the convention $\lambda_i = M$ for $i \leq \dim \ker \mathcal{L}$, with $M$ an abstraction representing $+\infty$, and $(\lambda_i)$ decreasing towards zero afterwards. This decomposition reads

$$\mathcal{L}^{-1} = \sum_{i \in \mathbb{N}} \lambda_i e_i \otimes e_i. \tag{16}$$

Note that the fact that all the $(\lambda_i)$ are positive, is due to the fact that $\mathcal{L}^{-1}$ is the inverse of a positive self-adjoint operator. As a consequence, the diffusion operator has discrete spectrum, and can be written as

$$\mathcal{L} = \sum_{i \in \mathbb{N}} \lambda_i^{-1} e_i \otimes e_i. \tag{17}$$

In such a setting, the kernel-free Tikhonov regularization Eq. (3) reads

$$g_\lambda = \sum_{i \in \mathbb{N}} \psi(\lambda_i) \left\langle g_\rho, \lambda_i^{1/2} e_i \right\rangle_{L^2} \lambda_i^{1/2} e_i, \tag{18}$$

with $\psi : x \rightarrow (x + \lambda)^{-1}$, and the convention $M\psi(M) = 1$.

## C.2 Regularity of the eigen vectors of the diffusion operator

In this subsection, we discuss on the regularity assumed in Assumption 2.

Introducing the kernel $k$ and its associated RKHS $\mathcal{H}$ is useful when the eigen vectors of $\mathcal{L}$ can be well approximated by functions in $\mathcal{H}$. In applications, people tends to go for kernels that are translation-invariant, which implied that the RKHS $\mathcal{H}$ is made of smooth functions, could it be analytical functions (for the Gaussian kernel) or functions in $H^m$ (for Sobolev kernels). As a consequence, we should investigate on the regularity of those eigen vectors. Indeed, if $\rho$ derives from a Gibbs potential, that is $\rho(\mathrm{d}x) = e^{-V(x)}\,\mathrm{d}x$, the eigen vectors of $\mathcal{L}$ can be shown to inherit from the smoothness of the potential $V$ [41]. For example if $V$ belongs to $H^m$, and $H^m \subset \mathcal{H}$, we expect $(e_i)$ to belongs to $\mathcal{H}$, thus verifying Assumption 2.

**Counter-example and beyond Assumption 2.** Note that if $\rho$ has several connected components of non-empty interiors, the null space of $\mathcal{L}$ is made of functions that are constants on each connected components of $\operatorname{supp}\rho_{\mathcal{X}}$. Those functions are not analytic. In such a setting, the Gaussian kernel is not sufficient for Assumption 2 to hold, and one should favor kernel associated with richer functional space such as the Laplace kernel or the neural tangent kernel [12]. However, as illustrated by Figure 2, $e_i$ not belonging to $\mathcal{H}$ does not mean that $e_i$ can not be well approximated by $\mathcal{H}$. Indeed it is well known that the approximation power of $\mathcal{H}$ for $e_i$ can be measure in the biggest power $p$ such that $e_i \in \operatorname{im} K^p$ [10], where $K = SS^*$. Assumption 2 corresponds to $p = 1/2$, but it should be seen as a specific instance of more generic approximation conditions.

**Handling constants in RKHS.** Finally, note also that many RKHS do not contain constant functions, and therefore might not contain the constant function $e_0$ (although we are only looking for equality in the support of $\rho_{\mathcal{X}}$), however this specific point with $e_0$ can easily be circumvent either by assuming that $g_\rho$ has zero mean, either by centering the covariance matrices $\Sigma$ and $\hat{\Sigma}$ [41]. This relates with the usual technique for SVM consisting in adding a unpenalized bias [49].

## C.3 Low-density separation

In this subsection, we discuss on how Assumption 1 relates to the idea of low-density separation.

**Low-variation intuition.** The low-density separation supposes that the variations of $g^*$ take place in region with low-density, so that $\left\|\mathcal{L}^{1/2}g^*\right\| / \|g^*\|$ is small. As such, using Courant-Fischer principle, Assumption 1 can be reformulated as $g^*$ belonging to the space

$$\operatorname{Span}\{e_i\}_{i\le r} = \underset{\substack{\mathcal{F}\subset L^2;\\ \dim\mathcal{F}=r}}{\arg\min} \max_{g\in\mathcal{F}} \frac{\left\|\mathcal{L}^{-1/2}f\right\|_{L^2}^2}{\|f\|_{L^2}^2}.$$

In other terms, Assumption 1 can be restated as $g^*$ belonging to a finite dimensionnal space that minimizes a measure of variation given by the Dirichlet energy.

To tell the story differently, suppose that we are in a classification setting, *i.e.* $Y \in \{-1, 1\}$, and that the $\operatorname{supp}\rho_{\mathcal{X}}$ is connected. Then we know that the null space of $\mathcal{L}$ is made of constant functions. Then the first eigen vector $e_2$ of $\mathcal{L}$ is a function that is orthogonal to constants. Hence $e_2$ is a function that changes its sign and that is "balanced" in the sense that $\mathbb{E}[e_2] = 0$ – *i.e.* if $e_2(x) = \mathbb{E}_\mu[Y \mid X = x]$ for some measure $\mu$, we have $\mathbb{E}_\mu[Y] = 0$, meaning that classes are "balanced". Moreover, in order to minimize $\left\|\mathcal{L}^{1/2}e_2\right\|$, the variations of $e_2$ should take place in low-density regions of $\mathcal{X}$.

**Diffusion intuition.** Finally, as $\mathcal{L}$ is a diffusion operator, we also have an interpretation of Assumption 1 is term of diffusion. Consider $(\lambda_i, e_i)$ the eigen elements of Eq. (17). The diffusion of $g_\rho$ according the density $\rho_{\mathcal{X}}$ can be written as, for $t \in \mathbb{R}$,

$$g_t = e^{-t\mathcal{L}}g_\rho = \sum_{i\in\mathbb{N}} e^{-t\lambda_i^{-1}} \langle g_\rho, e_i\rangle\, e_i.$$

This diffusion will cut off the high frequencies of $g_\rho$ that corresponds to $\langle g_\rho, e_i\rangle$ for big $i$, and big $\lambda_i^{-1}$. Indeed, the difference between the diffusion and the original $g_\rho$ can be measured as

$$\|g_t - g_\rho\|_{L^2}^2 = \sum_{i\in\mathbb{N}} (e^{-t\lambda_i^{-1}} - 1)^2 \langle g_\rho, e_i\rangle^2 = \sum_{i\in\mathbb{N}} t^2\lambda_i^{-2} \langle g_\rho, e_i\rangle^2 + o(t^2\lambda_i^{-2}).$$

So that assuming that $g_\rho$ is supported on few of the first eigen vectors of $\mathcal{L}$, can be rephrased as saying that the diffusion of $g_\rho$ does not modify it too much.

**The variance $\sigma_\ell$.** Theorem 1 shows that the need for labels depends on the variance parameter $\sigma_\ell^2$. It is natural to wonder on how this parameters relates to the low-density hypothesis. As we discussed, this parameter is linked to the variance of $Z = Y(I + \lambda\mathcal{L})^{-1}\delta_X$. We can separate the variability of this variable due to $X$ and the variability due to $Y$

$$Z = Z_X + Z_Y, \qquad \text{with} \qquad Z_X = (I + \lambda\mathcal{L})^{-1}g_\rho(X)\delta_X, \quad Z_Y = (I + \lambda\mathcal{L})^{-1}(Y - \mathbb{E}[Y \mid X])\delta_X.$$

As such we see that this variance depends on the structure of the density $\rho_{\mathcal{X}}$ with the variance of $(I + \lambda\mathcal{L})^{-1}\delta_X$, and the labelling noise with the variance of $(Y \mid X)$. The low-density separation does not tell us anything about the level of noise in $Y$ or the diffusion structure linked with $\rho_{\mathcal{X}}$, but additional hypothesis could be made to characterize those.

### C.4 Kernel operators

In this subsection, we define formally the operators $S$ and $\Sigma$.

We now turn towards operators linked with the Hilbert space $\mathcal{H}$. Recall that for $k : \mathcal{X} \to \mathcal{X} \to \mathbb{R}$ a kernel, $\mathcal{H}$ is defined the closure of the span of the elements $k_x$ under the scalar product $\langle k_x, k_{x'} \rangle = k(x, x')$. In particular, $\|k_x\|_{\mathcal{H}}^2 = k(x, x)$. $\mathcal{H}$ parameterize a vast class of function in $\mathbb{R}^{\mathcal{X}}$ through the mapping

$$S : \quad \mathcal{H} \quad \to \quad \mathbb{R}^{\mathcal{X}}$$
$$\theta \quad \to \quad (\langle k_x, \theta \rangle)_{x \in \mathcal{X}}.$$

Under mild assupmtions, $S$ maps $\mathcal{H}$ to a space of function belongs to $L^2$.

**Proposition 2.** *When $x \to k(x, x)$ belongs to $L^1(\rho_{\mathcal{X}})$, $S$ is a continuous mapping from $\mathcal{H}$ to $L^2(\rho_{\mathcal{X}})$. This is particularly the case when $\rho_{\mathcal{X}}$ has compact support and $k$ is continuous.*

*Proof.* Consider $\theta \in \mathcal{H}$, we have

$$\|S\theta\|_{L^2}^2 = \int_{\mathcal{X}} \langle k_x, \theta \rangle^2 \rho_{\mathcal{X}}(\mathrm{d}x) \le \int_{\mathcal{X}} \langle k_x, \theta \rangle_{\mathcal{H}}^2 \rho_{\mathcal{X}}(\mathrm{d}x) \le \int_{\mathcal{X}} \|k_x\|_{\mathcal{H}}^2 \|\theta\|_{\mathcal{H}}^2 \rho_{\mathcal{X}}(\mathrm{d}x)$$

$$= \|\theta\|_{\mathcal{H}}^2 \int_{\mathcal{X}} k(x, x)\rho_{\mathcal{X}}(\mathrm{d}x) = \|\theta\|_{\mathcal{H}}^2 \|x \to k(x, x)\|_{L^1}.$$

Moreover, when $\rho_{\mathcal{X}}$ has compact support and $k$ is continuous, $k$ is bounded on the support of $\rho_{\mathcal{X}}$ therefore $x \to k(x, x)$ belongs to $L^1$. $\qquad\square$

As a continuous operator from the Hilbert space $\mathcal{H}$ to the Hilbert space $L^2$, $S$ is naturally associated with many linear structure. In particular its adjoint $S^\star$, but also the self-adjoint operators $K = SS^\star$ and $\Sigma = S^\star S$.

**Proposition 3.** *The adjoint of $S$ is defined as*
$$S^\star : \quad L^2 \quad \to \quad \mathcal{H}$$
$$g \quad \to \quad \int_{\mathcal{X}} g(x)k_x \rho_{\mathcal{X}}(\mathrm{d}x) = \mathbb{E}_{X \sim \rho_{\mathcal{X}}}[g(X)k_X].$$
*To $S$ is associated the kernel self-adjoint operator on $L^2$*
$$K := SS^\star : \quad L^2 \quad \to \quad L^2$$
$$g \quad \to \quad (x \to \int_{\mathcal{X}} k(x, x')g(x')\rho_{\mathcal{X}}(\mathrm{d}x')),$$
*as well as the (not-centered) covariance on $\mathcal{H}$, $\Sigma := S^\star S = \mathbb{E}_{X \sim \rho_{\mathcal{X}}}[k_X \otimes k_X].$*

*Proof.* We shall prove the equality defining those operators. Consider $\theta \in \mathcal{H}$ and $g \in L^2$, we have

$$\langle S^\star g, \theta \rangle_{\mathcal{H}} = \langle g, S\theta \rangle_{L^2} = \mathbb{E}_{X \sim \rho_{\mathcal{X}}}[g(X) \langle k_X, \theta \rangle_{\mathcal{H}}] = \langle \mathbb{E}_{X \sim \rho_{\mathcal{X}}}[g(X)k_X], \theta \rangle_{\mathcal{H}}.$$

We also have, for $x \in \mathcal{X}$,

$$(SS^\star g)(x) = \langle k_x, \mathbb{E}_{X \sim \rho_{\mathcal{X}}}[g(X)k_X] \rangle_{\mathcal{H}} = \mathbb{E}_{X \sim \rho_{\mathcal{X}}}[g(X) \langle k_x, k_X \rangle_{\mathcal{H}}] = \mathbb{E}_{X \sim \rho_{\mathcal{X}}}[g(X)k(X, x)].$$

Finally, we have

$$S^\star S\theta = \mathbb{E}_{X \sim \rho_{\mathcal{X}}}[S\theta(X)k_X] = \mathbb{E}_{X \sim \rho_{\mathcal{X}}}[\langle \theta, k_X \rangle_{\mathcal{H}} k_X] = \mathbb{E}_{X \sim \rho_{\mathcal{X}}}[k_X \otimes k_X]\theta.$$

This provides the last of all the equalities stated above. $\qquad\square$

**The functional space $\mathcal{H}$.** In the main text, we have written everything in term of $\theta$, highlighting the parametric nature of kernel methods. This made it easier to dissociate the norm on functions derived from $\mathcal{H}$ and the one derived from $L^2$ or $H^1$. In literature, people tends to keep everything in term of functions $g_\theta = S\theta$ without even mentioning the dependency in $\theta$. Such a setting consists in considering directly the RKHS $\mathcal{H}$ whose scalar product is defined for $g, g' \in (\ker K)^\perp$ by $\langle g, g' \rangle_\mathcal{H} = \langle g, K^{-1} g' \rangle_{L^2}$.

## C.5 Derivative operators

In this subsection, we discuss on derivative in RKHS and we define formally the operator $L$.

As well as evaluation maps can be represented in $\mathcal{H}$, under mild assumptions, derivative evaluation maps can be benefited of such a property. Indeed, for $g_\theta = S\theta$, $x \in \mathcal{X}$ and $u \in \mathcal{B}_\mathcal{X}(0, 1)$ a unit vector, we have

$$\partial_u g_\theta(x) = \lim_{t \to 0} \frac{g_\theta(x + tu) - g_\theta(x)}{t} = \lim_{t \to 0} \frac{\langle \theta, k_{x+tu} \rangle_\mathcal{H} - \langle g, k_x \rangle_\mathcal{H}}{t} = \lim_{t \to 0} \left\langle \theta, \frac{k_{x+tu} - k_x}{t} \right\rangle_\mathcal{H}$$

As a linear combination of elements in $\mathcal{H}$, the difference quotient evaluation map $t^{-1}(k_{x+tu} - k_x)$ belongs to $\mathcal{H}$ and has a norm

$$\left\| \frac{k_{x+tu} - k_x}{t} \right\|_\mathcal{H}^2 = \frac{k(x + tu, x + tu) - 2k(x + tu, x) + k(x, x)}{t^2}.$$

In order for the limit when $t$ goes to zero to belong to $\mathcal{H}$, we see the importance of $k$ to be twice differentiable. This limit $\partial_u k_x$, whose existence is proven formally by Zhou [61], provides a derivative evaluation map in the sense that

$$\partial_u g_\theta(x) = \langle \theta, \partial_u k_x \rangle_\mathcal{H}.$$

From this equality, we derive that $\partial_{1i} k(x, x') = \langle k_{x'}, \partial_i k_x \rangle$, and recursively that $\langle \partial_i k_x, \partial_j k_{x'} \rangle = \partial_{1i} \partial_{2j} k(x, x')$.

Similarly to the operator $S$, we can introduce the operators $Z_i$ for $i \in [1, d]$, defined as

$$Z_i : \quad \begin{array}{ccc} \mathcal{H} & \to & \mathbb{R}^\mathcal{X} \\ \theta & \to & (\langle \partial_i k_x, \theta \rangle_\mathcal{H})_{i \leq d} \end{array}.$$

Once again, under mild assumptions, $\operatorname{im} Z_i$ inherit from an Hilbertian structure.

**Proposition 4.** *When $x \to \partial_{1i} \partial_{2i} k(x, x)$ belongs to $L^1(\rho_\mathcal{X})$, $Z_i$ is a continuous mapping from $\mathcal{H}$ to $L^2(\rho_\mathcal{X})$. This is particularly the case when $\rho_\mathcal{X}$ has compact support and $k$ is twice differentiable with continuous derivatives.*

*Proof.* Consider $\theta \in \mathcal{H}$, similarly to before, we have

$$\|Z\theta\|_{L^2}^2 = \int_\mathcal{X} \langle \partial_i k_x, \theta \rangle^2 \rho_\mathcal{X}(\mathrm{d}x) \leq \|\theta\|_\mathcal{H}^2 \int_\mathcal{X} \|\partial_i k_x\|_\mathcal{H}^2 \rho_\mathcal{X}(\mathrm{d}x) = \|\theta\|_\mathcal{H}^2 \|x \to \partial_{1,i} \partial_{2,i} k(x, x)\|_{L^1}.$$

Moreover, when $\rho_\mathcal{X}$ has compact support and $\partial_{1,i} \partial_{2,i} k$ is continuous, $\partial_{1,i} \partial_{2,i} k$ is bounded on the support of $\rho_\mathcal{X}$ therefore $x \to \partial_{1,i} \partial_{2,i} k$ belongs to $L^1$. $\qquad \square$

Among the linear operator that can be build from $Z_i$, in the theoretical part of this paper, we are mainly interested in $Z_i^\star Z_i$. In the empirical part however, we might be interested in $Z_i Z_j^\star$ as well as $Z_i S^\star$ as it might appear in Algorithm 1 (where $Z_n$ has to be understood as the empirical version of $Z = [Z_1; \cdots; Z_d]$).

**Proposition 5.** *The energy Dirichlet on $\mathcal{H}$ can be represented through the operator*

$$S^\star \mathcal{L} S = \sum_{i=1}^d Z_i^\star Z_i = \sum_{i=1}^d \mathbb{E}_{X \sim \rho_\mathcal{X}}[\partial_i k_X \otimes \partial_i k_X].$$

*Proof.* Let $\theta \in \mathcal{H}$ and $g_\theta = S\theta$, we have

$$\langle g_\theta, \mathcal{L}g_\theta \rangle_{L^2} = \langle \theta, S^\star \mathcal{L} S\theta \rangle_{\mathcal{H}} = \mathbb{E}_{X \sim \rho_{\mathcal{X}}} \left[ \|\nabla g_\theta(X)\|^2 \right] = \sum_{i=1}^d \mathbb{E}_{X \sim \rho_{\mathcal{X}}} \left[ (\partial_i g_\theta(X))^2 \right]$$

$$= \sum_{i=1}^d \mathbb{E}_{X \sim \rho_{\mathcal{X}}} \left[ \langle \partial_i k_X, \theta \rangle_{\mathcal{H}}^2 \right] = \sum_{i=1}^d \|Z_i \theta\|_{L^2}^2 = \left\langle \theta, \sum_{i=1}^d Z_i^\star Z_i \theta \right\rangle_{\mathcal{H}}$$

$$= \sum_{i=1}^d \mathbb{E}_{X \sim \rho_{\mathcal{X}}} \left[ \langle \theta, (\partial_i k_X \otimes \partial_i k_X)\theta \rangle_{\mathcal{H}} \right] = \left\langle \theta, \sum_{i=1}^d \mathbb{E}_{X \sim \rho_{\mathcal{X}}} \left[ \partial_i k_X \otimes \partial_i k_X \right] \theta \right\rangle_{\mathcal{H}}.$$

Since the three operators are self-adjoint and they all represent the same quadratic form, they are equals. $\qquad\square$

## C.6 Relation between $\Sigma$ and $L$

In this subsection, we discuss on the relation between $\Sigma$ and $L$ and show that we can expect the existence of $a \in (1 - 2/d, 1]$ and $c > 0$ such that $L \preceq c\Sigma^a$.

**Informal capacity considerations.** We want to compare $\Sigma$ and $L$, as $L \preceq c\Sigma^a$ with the biggest $a$ possible. This depend on how fast the eigen values are decreasing, which is linked to the entropy numbers of those two compact operators. Those entropy numbers are linked with the capacity of the functional spaces $\left\{ g \in L^2 \,\middle|\, \left\| K^{-1/2} g \right\|_{L^2} < \infty \right\}$ and $\left\{ g \in L^2 \,\middle|\, \left\| K^{-1/2} \mathcal{L}^{-1/2} g \right\|_{L^2} < \infty \right\}$. The first space is the reproducing kernel Hilbert space linked with $k$, the second space is, roughly speaking, a space of function whose integral belongs to the first space. As such, if the first space is $\mathcal{H}^m$, the second is $\mathcal{H}^{m-1}$, and we can consider $a = (m-1)/m$. Because we are considering kernel, we have $m > d/2$ (this to make sure that the evaluation functionals $L_X : f \to f(x)$ are continuous), so that $a > 1 - 2/d$. Without trying to make those "algebraic" considerations more formal, we will give an example on the torus.

**Translation-invariant kernel and Fourier transform.** Consider $L^2([0,1]^d, \mathrm{d}x)$ the space of periodic functions in dimension $d$, square integrable against the Lebesgue measure on $[0,1]^d$. For simplicity, we will suppose that $\rho_{\mathcal{X}}$ is the Lebesgue measure on $[0,1]^d$. Consider a translation invariant kernel

$$k(x,y) = q(x - y) \qquad \text{for } q : \mathbb{R}^d \to \mathbb{R} \text{ that is one periodic.}$$

In this setting, the operator $K$, operating on $L^2$, is the convolution by $q$, that is

$$K : \begin{array}{ccc} L^2 & \to & L^2 \\ g & \to & q * g \end{array}, \qquad \text{hence} \qquad \widehat{Kg} = \hat{q}\hat{g}.$$

Where we have used the fact that convolutions can be represented by a product in Fourier. Note that, from Böchner theorem, we know that $k$ being positive definite implies that the Fourier transform of $q$ exists and is not negative. Let us define the Fourier coefficient and the inverse Fourier transform as

$$\forall \omega \in \mathbb{Z}^d, \quad \hat{g}(\omega) = \int_{[0,1]^d} g(x) e^{-2i\pi \omega^\top x} \, \mathrm{d}x, \quad \text{and} \quad \forall x \in [0,1]^d, \quad g(x) = \sum_{\omega \in \mathbb{Z}^d} e^{2i\pi \omega^\top x} \hat{g}(\omega).$$

$K$ being a convolution operator, it is diagonalizable with eigen elements $(\hat{q}(\omega), x \to e^{2i\pi \omega^\top x})_{\omega \in \mathbb{Z}^d}$. From this, we can explicit many of our abstract operators. First of all, using Perceval's theorem,

$$\|g\|_{\mathcal{H}}^2 = \left\langle g, K^{-1} g \right\rangle_{L^2} = \sum_{\omega \in \mathbb{Z}^d} \frac{|\hat{g}(\omega)|^2}{\hat{q}(\omega)}.$$

Hence we can parametrize $\mathcal{H}$ with $(\theta_\omega)_{\omega \in \mathbb{Z}^d} \in \mathbb{C}^{\mathbb{Z}^d}$ and the $\ell^2$-metric, where $\theta_\omega = \hat{g}(\omega)/\sqrt{\hat{q}(\omega)}$ and

$$(S\theta)(x) = g_\theta(x) = \sum_{\omega \in \mathbb{Z}^d} e^{2i\pi \omega^\top x} \sqrt{\hat{q}(\omega)} \theta_\omega.$$

Note that this is not the usual parameterization of $\mathcal{H}$ by elements $\theta \in \mathcal{H}$ as $(\mathbb{C}^{\mathbb{Z}^d}, \ell^2)$ is not a space of functions. However, such a parametrization of $\mathcal{H}$ does not change any of the precedent algebraic considerations on the operators $S$, $\Sigma$, $K$, and $L$.

**Diffusion operator and Fourier transform.** As well as convolution operators are well represented in Fourier, derivation operators are. Indeed, when $g$ is regular, we have

$$\left\| \mathcal{L}^{1/2} g \right\|_{L^2}^2 = \|\nabla g\|_{L^2}^2 = \sum_{j=1}^d \|\partial_j g\|_{L^2}^2 = \sum_{j=1}^d \sum_{\omega \in \mathbb{Z}^d} \omega_j^2 \, |\hat{g}(\omega)|^2.$$

As a consequence, using the expression of $S\theta$, we have

$$\Sigma \theta = \sum_{\omega \in \mathbb{Z}^d} \hat{q}(\omega) \theta_\omega, \qquad \text{while} \qquad L\theta = \sum_{\omega \in \mathbb{Z}^d} \|\omega\|_2^2 \, \hat{q}(\omega) \theta_\omega, \quad \text{where} \quad \|\omega\|_2^2 = \sum_{j=1}^d \omega_i^2.$$

With this parameterization, the eigen elements of $\Sigma$ are $(\hat{q}(\omega), \delta_\omega)_{\omega \in \mathbb{Z}^d}$ while the one of $L$ are $(\|\omega\|_2^2 \, \hat{q}(\omega), \delta_\omega)_{\omega \in \mathbb{Z}^d}$.

**Eigen value decay comparison.** Hence, having $L \preceq c\Sigma^a$ is equivalent to having $\|\omega\|_2^2 \, \hat{q}(\omega) \le c\hat{q}(\omega)^a$. Now suppose that the decay of $\hat{q}$ is governed by

$$c_1 (1 + \sigma^{-1} \|\omega\|_2^2)^{-m} \le \hat{q}(\omega) \le c_2 (1 + \sigma^{-1} \|\omega\|_2^2)^{-m},$$

for two constants $c_1, c_2 > 0$. In particular, this is verified for Matérn kernels, corresponding to the fractional Sobolev space $H^m$, and for the Laplace kernel with $m = (d+1)/2$, which reads $k(x, y) = \exp(-\sigma^{-1} \|x - y\|)$. The Gaussian kernel could be seen as $m = +\infty$ as it has exponential decay. With such a decay we have, assuming without restrictions that we are in one dimension

$$\omega^2 \hat{q}(\omega) \le c_2 \omega^2 (1 + \sigma^{-1}\omega^2)^{-m} \le c_2 \sigma (1 + \sigma^{-1}\omega^2)^{-(m-1)} \le c_1^{\frac{m}{m-1}} c_2 \sigma \hat{q}(\omega)^{\frac{m-1}{m}}.$$

In other terms, we can consider $c = c_1^{\frac{m}{m-1}} c_2 \sigma$ and $a = (m-1)/m$. Assuming that $q$ is square-integrable, so is $\hat{q}$, which implies that $2m > d$. As a consequence, we do have $a > 1 - 2/d$. Note that this reasoning could be extended to the case where $\rho_{\mathcal{X}}$ has a density against the Lebesgue measure, that is bounded above and below away from zero.

## D  Spectral decomposition

In this section, we recall facts on spectral regularization, before proving Proposition 1 and extending it to the case $\mu = 0$.

### D.1  Generalized singular value with matrices

**Generalized singular value decomposition.** Let $A \in \mathbb{R}^{m_1 \times n}$ and $B \in \mathbb{R}^{m_2 \times n}$ be two matrices. There exists $U \in \mathbb{R}^{m_1 \times m_1}$, $V \in \mathbb{R}^{m_2 \times m_2}$ two orthogonal matrices, $c \in \mathbb{R}^{m_1 \times r}$ and $s \in \mathbb{R}^{m_2 \times r}$ be two 1-diagonal matrices such that $c^\top c + s^\top s = I_r$, and $H \in \mathbb{R}^{n \times r}$ a non-singular matrix such that

$$A = UcH^{-1}, \qquad B = VsH^{-1}.$$

To be more precise $c$ is such that only entries $c_{ii} = \cos(\theta_i)$ for $i < \min(r, m_2)$ are non-zeros and $s$ such that only entries $s_{m_1-i, r-i} = \sin(\theta_{r-i})$ for $i < \min(r, m_1)$ are non-zeros, with $\theta_i \in [-\pi/2, \pi/2]$ an angle. Here, $c$ stands for cosine, $s$ for sinus and $r$ for rank.

**Link with generalized eigenvalue problem.** As well as the singular value of $A$ is linked with the eigen value of $A^\top A$, the generalized singular value decomposition of $[A; B]$ is linked with the generalized eigenvalue problem linked with $(A^\top A, B^\top B)$. Indeed, we have

$$A^\top A = H^{-\top} c^\top c H^{-1}, \qquad B^\top B = H^{-\top} s^\top s H^{-1}.$$

In particular, with $(e_i)$ the canonical basis of $\mathbb{R}^r$, and $h_i$ the $i$-th column of $H$, we get

$$H^\top A^\top A h_i = \cos(\theta_i)^2 e_i = \tan(\theta_i)^{-2} \sin(\theta_i)^2 e_i = \tan(\theta_i)^{-2} H^\top B^\top B h_i.$$

From which we deduce that, since $\operatorname{im} A \cup \operatorname{im} B \subset \operatorname{im} H^\top$,

$$A^\top A h_i = \tan(\theta_i)^{-2} B^\top B h_i, \qquad h_j^\top B^\top B h_i = \sin(\theta_i)^2 \mathbf{1}_{i=j}.$$

So if we denote by $f_i = |\sin(\theta_i)|^{-1} h_i$ and $\lambda_i = \tan(\theta_i)^{-2}$, assuming $\lambda_i \ne 0$ for all $i \le r$ (which corresponds to $\ker B \subset \ker A$), $(\lambda_i)_{i \le r}, (f_i)_{i \le r}$ provide the generalized eigenvalue decomposition of $(A^\top A, B^\top B)$ in the sense that

$$A^\top A f_i = \lambda_i B^\top B f_i, \qquad f_j^\top B^\top B f_i = \mathbf{1}_{i=j}, \qquad f_j^\top A^\top A f_i = \lambda_i \mathbf{1}_{i=j}.$$

## D.2   Tikhonov regularization

Define the Tikhonov regularization

$$x_\lambda = \arg\min_{x \in \mathbb{R}^n} \|Ax - b\|^2 + \lambda \|Bx\|^2 .$$

When this problem is well-defined, the solution is defined as

$$x_\lambda = (A^\top A + \lambda B^\top B)^\dagger A^\top b.$$

With the generalized singular value decomposition of $A$ and $B$, we have

$$A^\top A + \lambda B^\top B = H^{-\top} \gamma_\lambda H^{-1}, \quad \text{with} \quad \gamma_\lambda = c^\top c + \lambda s^\top s.$$

Using the fact that $A^\top b = H^{-\top} c^\top U^\top b$, we get

$$x_\lambda = H \gamma_\lambda^{-1} c^\top U^\top b = \left( \sum_{i=1}^r \frac{\cos(\theta_i)}{\cos(\theta_i)^2 + \lambda \sin(\theta_i)^2} h_i \otimes u_i \right) b.$$

Now, we would like to replace $c_{ii}$, $s_{ii}$, $h_i$ and $u_i$ with quantities that depend on $\lambda_i$, $f_i$ and $A$. To do so recall that $AH = Uc$, therefore $\cos(\theta_i) u_i = Ah_i$, and recall that $h_i = \sin(\theta_i) f_i$ and $\lambda_i = \cos(\theta_i)^2 / \sin(\theta_i)^2$. Inputting those equality in the last expression of $x_\lambda$ we get

$$x_\lambda = \left( \sum_{i=1}^r \frac{\sin(\theta_i)^2}{\cos(\theta_i)^2 + \lambda \sin(\theta_i)^2} f_i \otimes Af_i \right) b = \left( \sum_{i=1}^r \frac{1}{\lambda_i + \lambda} f_i \otimes Af_i \right) b.$$

Finally,

$$b_\lambda = Ax_\lambda = \sum_{i=1}^r \psi(\lambda_i) \langle Af_i, b \rangle Af_i, \qquad \text{where} \qquad \psi(x) = \frac{1}{x + \lambda}.$$

## D.3   Extension to operators

To end the proof of Proposition 1, we should prove that we can apply the generalized eigenvalue decomposition to operators. We will only prove that it is possible for $(\Sigma, L + \mu)$ based on simple considerations.

**Proposition 6.** *When $k$ is continuous and* $\operatorname{supp} \rho_\mathcal{X}$ *is bounded, $\Sigma$ is a compact operator.*

*Proof.* We have $\Sigma = \mathbb{E}[k_X \otimes k_X]$ and $\|k_x\| = k(x,x)$. Since $k$ is continuous and $\operatorname{supp} \rho_\mathcal{X}$ is compact, for $x \in \operatorname{supp} \rho_\mathcal{X}$, $k(x,x)$ is bounded. Hence $\Sigma$ is a nuclear operator, hence trace class and compact. $\square$

**Proposition 7.** *When $k$ is twice differentiable with continuous derivative, and* $\operatorname{supp} \rho_\mathcal{X}$ *is compact, $L$ is a compact operator. As a consequence, $L$ has a compact spectrum, and has a pseudo inverse that we will denote, with a slight abuse of notation, by $L^{-1}$.*

*Proof.* The proof is similar to the one showing that $\Sigma$ is compact, based on the fact that $L = \sum_{i=1}^d \mathbb{E}[\partial_i k_X \otimes \partial_i k_X]$, and $\|\partial_i k_X\|^2 = \partial_{1i} \partial_{2i} k(x,x)$. $\square$

**Proposition 8.** *When $\Sigma$ is compact, for all $\mu > 0$, $(L+\mu)^{-1/2} \Sigma (L+\mu)^{-1/2}$ is a compact operator.*

*Proof.* The proof is straightforward

$$\operatorname{Tr}((L+\mu)^{-1/2} \Sigma (L+\mu)^{-1/2}) = \operatorname{Tr}(\Sigma (L+\mu)^{-1}) \le \left\| (L+\mu)^{-1} \right\|_{\mathrm{op}} \operatorname{Tr}(\Sigma) \le \mu^{-1} \operatorname{Tr}(\Sigma) < +\infty.$$

Therefore the operator is trace class, hence compact. $\square$

**Proposition 9.** *For any $\mu > 0$, the generalized eigen value decomposition of $(\Sigma, L + \mu)$ as defined in Proposition 1 exists.*

*Proof.* Using the spectral theorem, since $(L + \mu)^{-1/2}\Sigma(L + \mu)^{-1/2}$ is positive self-adjoint compact operator, there exists $(\xi_i)$ a basis of $\mathcal{H}$ and $(\lambda_i) \in \mathbb{R}_+$ a decreasing sequence (note that $\ker(L + mu) = \ker \Sigma = \{0\}$), such that

$$(L + \mu)^{-1/2}\Sigma(L + \mu)^{-1/2} = \sum_{i \in \mathbb{N}} \lambda_i \xi_i \otimes \xi_i.$$

Taking $\theta_i = (L + \mu)^{-1/2}\xi_i$, we get $\Sigma\theta_i = \lambda_i L\theta_i$. Because $(\xi_i)$ generates $\mathcal{H}$, and $(L + \mu)^{-1/2}$ is bijective (since $L$ is compact, $(L + \mu)^{-1}$ is coercive), $((L + \mu)^{-1/2}\xi_i)$ generates $\mathcal{H}$. $\qquad\square$

Proposition 1 follows from prior discussion on Tikhonov regularization extended to infinite summations.

## D.4 The case $\mu = 0$

When $\mu = 0$, Eq. (7) should be seen as the rewritting of Eq. (18) based on the RKHS $\mathcal{G} = \operatorname{im} S$. This can only be done when the eigen vectors of $\mathcal{L}$ appearing in Eq. (17) belongs to $\mathcal{G} = \operatorname{im} S = \operatorname{im} K^{1/2}$, which is exactly what Assumption 2 provides. In such a setting, we can find $(\theta_i) \in \mathcal{H}^{\mathbb{N}}$ to write $\lambda_i^{1/2}e_i = S\theta_i$ as soon as $\lambda_i \neq 0$ (write $Me_i = S\theta_i$ for $M$ an abstraction representing $+\infty$ when $\lambda_i = 0$, handling the potential fact that $\ker B \not\subset \ker A$), we get $\theta_i S^* S\theta_j = \lambda_i \mathbf{1}_{i=j}$, and $L\theta_i = \lambda_i^{-1}\Sigma\theta_i$, and we can extend Proposition 1 to the case $\mu = 0$, with

$$g_\lambda = \sum_{i \in \mathbb{N}} \psi(\lambda_i) \langle S^\star g_\rho, \theta_i \rangle S\theta_i, \tag{19}$$

where we handle the null space of $\mathcal{L}$ with the equality $M\psi(M) = 1$, verified by $M$ our abstraction representing $+\infty$, so that $\psi(M) \langle S^\star g_\rho, \theta_i \rangle S\theta_i = \langle g_\rho, e_i \rangle e_i$ as soon as $\lambda_i = 0$.

**Beyond Assumption 2.** Assumption 2 could be made generic by considering the biggest $(p_i) \in \mathbb{R}_+^{\mathbb{N}}$ such that $K^{-p_i}e_i$ belongs to $L^2$, and rewriting Eq. (19) under the form $g_\lambda = \sum_{i \in \mathbb{N}} \psi(\lambda_i) \langle (S_0 K^{p_i})^\star g_\rho, \theta_i \rangle S_0 K^{p_i}\theta_i$, with $\theta_i = \lambda_i^{-1/2}S_0^{-1}K^{-p_i}\theta_i$ and $S_0 = K^{-1/2}S$ the isomorphism between $\mathcal{H}$ and $L^2$ (assuming that $S$ is dense in $L^2$). Such an assumption would describe all situations from no assumption ($p_i = 0$ for all $i$), Assumption 2 ($p_i = 1/2$ for all $i$) to even more optimistic assumptions ($p_i \geq 1$ for all $i$).

# E  Consistency analysis

This section is devoted to the proof of Theorem 1. The proof is based on Eqs. (7) and (19), and splits the error of $\|g_\rho - \hat{g}_p\|_{L^2}^2$ into several components linked with how well we approximate $S^\star g_\rho$, and how well we approximate the eigenvalue decomposition $(\lambda_i, \theta_i)$ of $(\Sigma, L)$.

## E.1  Sketch and understanding of the proof

In this subsection, we explain how work the proofs for consistency theorems such as Theorem 1.

Let us define the mapping $G : \mathcal{H} \times \mathcal{C} \to L^2$ with $\mathcal{C}$ the set of pairs of self-adjoint operators on $\mathcal{H}$ that admit a generalized eigen value decomposition, as

$$G(\theta, (A, B)) = \sum_{i \in \mathbb{N}} \psi(\lambda_i) \langle \theta, \theta_i \rangle S\theta_i \qquad \text{with} \qquad (\lambda_i, \theta_i) \in \text{GEVD}(A, B). \tag{20}$$

$G(\theta, (A, B)) \in L^2$ corresponds to writing $\theta \in \mathcal{H}$ in the basis associated with the generalized eigen value decomposition (GEVD) of $(A, B)$.

**Proposition 10.** *Under Assumptions 1 and 2, and with $\psi$ defined in Theorem 1*

$$g_\lambda = G(S^\star g_\rho, (\Sigma, L)), \qquad \text{and} \qquad \hat{g}_p = G(S^{\hat\star}g_\rho, (P\Sigma P, P\hat{L}P + \mu P)),$$

*with $P$ the projection matrix from $\mathcal{H}$ to $\text{Span}\{k_{X_i}\}_{i \leq p}$.*

*Proof.* This is a direct application of Assumptions 1, 2, Eq. (19) and Algorithm 1. □

The main point of the proof is to relate $g_\rho$ to $\hat{g}_p$. To do so, we will use several functions in $L^2$ generated by $G$. We detail our steps in Table 2. Table 2 gives a first answer to the two questions asked in the opening of Section 4. The number of unlabelled data controls the convergence of the operators $(P\hat{\Sigma}P, P\hat{L}P + \mu P)$ towards $(\Sigma, L + \mu)$. The number of labelled data controls the convergence of the vector $\widehat{S^\star g_\rho}$ towards $S^\star g_\rho$. Priors on the structure of the problem, such as source and approximation conditions, control the convergence of the bias estimate $g_{\lambda,\mu}$ towards $g_\rho$. Furthermore, a more precise study reveals that the concentration of operators are related to efficient dimension [10] and are accelerated by capacity assumptions on the functional space whose norm is $\|g\| = \left\|(\mathcal{L} + K^{-1})^{1/2}g\right\|_{L^2}$, and that the concentration of the vector $\widehat{S^\star g_\rho}$ is accelerated by assumptions on moments of the variable $Y(I + \lambda\mathcal{L})^{-1}\delta_X$ (inheriting randomness from $(X,Y) \sim \rho$).

**Table 2:** Steps in the consistency analysis

| Estimate | Vector | Property of convergence | Basis |
|---|---|---|---|
| $\hat{g}_p$ | $\widehat{S^\star g_\rho}$ | | $(P\hat{\Sigma}P, P\hat{L}P + \mu P)$ |
| | | Low-rank approximation [45] | $\downarrow$ |
| $\hat{g}$ | $\widehat{S^\star g_\rho}$ | | $(\hat{\Sigma}, \hat{L} + \mu)$ |
| | | Concentration for self-adjoint operators [36] | $\downarrow$ |
| $g_{n_\ell}$ | $\widehat{S^\star g_\rho}$ | | $(\Sigma, L + \mu)$ |
| | | Concentration for vector in Hilbert space [58] | $\downarrow$ |
| $g_{\lambda,\mu}$ | $S^\star g_\rho$ | | $(\Sigma, L + \mu)$ |
| | | Bias controlled with source condition [10, 30] | $\downarrow$ |
| $g_\lambda$ | $S^\star g_\rho$ | | $(\Sigma, L)$ |
| $\downarrow$ | | Bias controlled with source condition [10, 30] | |
| $g_\rho$ | | | |

**Control of biases.** We begin our study in a downward fashion regarding Table 2. Indeed, for Tikhonov regularization Eq. (3), we can show that for $q \in [0, 1]$,

$$\|g_\lambda - g_\rho\|_{L^2} \le \lambda^q \|\mathcal{L}^q g_\rho\|_{L^2}.$$

Meaning that if we have the source condition $g_\rho \in \text{im } \mathcal{L}^q$ (which is a condition on how fast $(\langle g_\rho, e_i \rangle)_{i \in \mathbb{N}}$ decreases compared to $(\lambda_i)_{i \in \mathbb{N}}$ for $(\lambda_i, e_i)$ the eigen value decomposition of $\mathcal{L}^{-1}$), the rates of convergence of this term when $n$ goes to infinity is controlled by the regularization scheme $\lambda_n^q$.

In a similar fashion to the kernel-free bias above, for $(q_i) \in (0,1)^{\mathbb{N}}$, we can have

$$\|g_{\lambda,\mu} - g_\lambda\|_{L^2}^2 \le 2 \sum_{i \in \mathbb{N}} \lambda^{2q_i} \mu^{2q_i} \left(\frac{\lambda_i}{\lambda + \lambda_i}\right)^2 |\langle e_i, g_\rho \rangle|^2 \left\|K^{-q_i} e_i\right\|_{L^2}^2.$$

This shows explicitly the usefulness of controlling at the same time how $g_\rho$ is supported on the eigen spaces of $\mathcal{L}$ and how the eigen vectors are well approximated by the RKHS $\mathcal{H}$, which can be read in the value of $(q_i)$ such that all $e_i \in \text{im } K^{q_i}$.

**Vector concentration.** Let us now switch to concentration of $\widehat{S^\star g_\rho} = n_\ell^{-1} \sum_{i=1}^{n_\ell} Y_i k_{X_i}$ towards $S^\star g_\rho = \mathbb{E}_{(X,Y)\sim\rho}[Yk_X]$, it will allow to control $\|g_{n_\ell} - g_{\lambda,\mu}\|_{L^2}^2$ with the notations appearing in Table 2. Note how this convergence should be measured in term of the reconstruction error

$$\left\| \sum_{i\in\mathbb{N}} \psi(\lambda_{i,\mu}) \left\langle S^\star g_\rho - \widehat{S^\star g_\rho}, \theta_{i,\mu} \right\rangle S\theta_{i,\mu} \right\|_{L^2}.$$

This error might behave in a must better fashion than the $L^2$ error between $SS^\star g_\rho$ and $S\widehat{S^\star g_\rho}$. In particular, on Figure 1, we can consider $\psi(\lambda_{i,\mu}) = 0$ for $i > 4$, and we might have $\langle Yk_X, \theta_{i,\mu} \rangle = Y\mathbf{1}_{x\in C_i}$, for $i \leq 4$ and $(X,Y) \in \mathcal{X} \times \mathcal{Y}$, where $C_i$ is the $i$-th innermost circle. In this setting, when all four $(Y \mid X \in C_i)$ are deterministic, we only need one labelled point per circle to clear the reconstruction error. Based on concentration results in Hilbert space, when $|Y|$ is bounded by a constant $c_\mathcal{Y}$, and $x \to k(x,x)$ by a constant $\kappa^2$, we have, with $\mathcal{D}_{n_\ell} \sim \rho^{\otimes n_\ell}$ the dataset generating the labelled data

$$\mathbb{E}_{\mathcal{D}_{n_\ell}}\left[ \|g_{n_\ell} - g_{\lambda,\mu}\|_{L^2}^2 \right] \leq 2\sigma_\ell^2(\mu\lambda n_\ell)^{-1} + \frac{4}{9}c_\mathcal{Y}^2\kappa^2(\mu\lambda n_\ell^2)^{-1}.$$

where $\sigma_\ell^2 \leq c_\mathcal{Y}^2 \operatorname{Tr}(\Sigma)$ is a variance parameter to relate to the variance of $Y(I + \lambda\mathcal{L})^{-1}\delta_X$ (where the randomness is inherited from $(X,Y) \sim \rho$). The fact that the need for labelled data depends on the variance of $(X,Y)$ after being diffused through $\mathcal{L}$ is coherent with the results obtained by Lelarge and Miolane [29] in the specific case of a mixture of two Gaussians.

**Basis concentration.** We are left with the comparison of $g_{n_\ell}$, which is the filtering of $\widehat{S^\star g_\rho}$ with the operators $(\Sigma, L + \mu)$, and $\hat{g}_p$, which is the filtering of the same vector with the operators $(P\hat{\Sigma}P, P\hat{L}P + \mu P)$. As the number of samples grows towards infinity, we know that $(P\hat{\Sigma}P, P\hat{L}P + \mu P)$ will converge in operator norm towards $(\Sigma, L + \mu)$. Yet, how to leverage this property to quantify the convergence of $\hat{g}_p$ towards $g_{n_\ell}$? Let us write $(\lambda_i, \theta_i) = \operatorname{GEVD}(\Sigma, L + \mu)$, and $(\lambda_i', \theta_i') = \operatorname{GEVD}(P\hat{\Sigma}P, P\hat{L}P + \mu P)$, we have

$$\|\hat{g}_p - g_{n_\ell}\|_{L^2} = \left\| \sum_{i\in\mathbb{N}} \psi(\lambda_i) \left\langle \theta_i, \hat{\theta}_\rho \right\rangle S\theta_i - \psi(\lambda_i') \left\langle \theta_i', \hat{\theta}_\rho \right\rangle S\theta_i' \right\| \qquad \text{with} \qquad \hat{\theta}_\rho = \widehat{S^\star g_\rho}.$$

The generic study of this quantity requires to control eigenspaces one by one. Note that we expect convergence of eigenspaces to depend on gaps between eigenvalues. However, when considering Tikhonov regularization, this quantity can be written under a simpler form. In particular, the concentration of operators is controlled, up to few leftovers, through the quantity $\left\| (\Sigma + \lambda L + \mu\lambda)^{-1/2}((\Sigma - \hat{\Sigma}) + \lambda(L - \hat{L}))(\Sigma + \lambda L + \mu\lambda)^{-1/2} \right\|_{\mathrm{op}}$ where $\|\cdot\|_{\mathrm{op}}$ designs the operator norm. In this setting, the low-rank approximation is controlled through $\left\| (\Sigma + \lambda L)^{1/2}(I - P) \right\|_{\mathrm{op}}$, and when $L \preceq c\Sigma^\alpha$, this term can be controlled by $\left\| \Sigma^{1/2}(I - P) \right\|_{\mathrm{op}} + \lambda^{1/2} \left\| \Sigma^{1/2}(I - P) \right\|_{\mathrm{op}}^\alpha$ which can be controlled based on the work of Rudi et al. [45].

## E.2 Risk decomposition

In this subsection, we decompose the risk appearing in Theorem 1.

### E.2.1 Control of biases

We begin by splitting the error $\|g_\rho - \hat{g}_p\|_{L^2}$ between a bias term due to the regularization parameters and a variance term due to the data. With the notation of Table 2,

$$\|g_\rho - \hat{g}_p\|_{L^2} \leq \|g_\rho - g_\lambda\|_{L^2} + \|g_\lambda - g_{\lambda,\mu}\|_{L^2} + \|g_{\lambda,\mu} - \hat{g}_p\|_{L^2}. \tag{21}$$

We will control the first two terms here, and the last term in the following subsections.

**Proposition 11** (Bias in $\lambda$). *Under Assumption 1*

$$\|g_\lambda - g_\rho\|_{L^2} \leq \lambda \|\mathcal{L}g_\rho\|_{L^2}. \tag{22}$$

*Proof.* Based on the definition of $g_\lambda = (I + \lambda\mathcal{L})^{-1}g_\rho$, we have

$$g_\lambda - g_\rho = ((I + \lambda\mathcal{L})^{-1} - I)g_\rho = -\lambda\mathcal{L}g_\rho.$$

Because $g_\rho$ is supported on the first eigen vectors of the Laplacian (Assumption 1), we have $g_\rho = \sum_{i=1}^r \langle g_\rho, e_i \rangle\, e_i$, with $e_i$ the eigen vector of $\mathcal{L}$ appearing in Eq. (17), and

$$\|\mathcal{L}g_\rho\|_{L^2}^2 = \left\|\sum_{i=1}^r \lambda_i^{-1} \langle g_\rho, e_i \rangle\, e_i\right\|_{L^2}^2 = \sum_{i=1}^r \lambda_i^{-2} \langle g_\rho, e_i \rangle^2 \le \lambda_r^{-2} \|g_\rho\|_{L^2}^2 < +\infty.$$

This ends the proof of this proposition. $\qquad\square$

**Proposition 12** (Bias in $\mu$). *Under Assumptions 1 and 2, we have*

$$\|g_{\lambda,\mu} - g_\lambda\|_{L^2}^2 \le \lambda\mu c_a^2 \|g_\rho\|_{L^2}^2, \qquad with \qquad c_a^2 = \sum_{i=1}^r \left\|K^{-1/2}e_i\right\|_{L^2}^2 = \sum_{i=1}^r \|e_i\|_{\mathcal{H}}^2. \qquad (23)$$

*Proof.* Before diving into the proof, recall that the RKHS norm penalization can be written as $\|g\|_{\mathcal{H}} = \left\|K^{-1/2}g\right\|_{L^2}$. Using the fact that $A^{-1} - B^{-1} = A^{-1}(B - A)B^{-1}$, we have

$$\begin{aligned}
g_{\lambda,\mu} - g_\lambda &= ((I + \lambda\mathcal{L} + \mu\lambda K^{-1})^{-1} - (I + \lambda\mathcal{L})^{-1})g_\rho \\
&= -(I + \lambda\mathcal{L} + \mu\lambda K^{-1})^{-1}\lambda\mu K^{-1}(I + \lambda\mathcal{L})^{-1}g_\rho \\
&= -(\lambda\mu)^{1/2}(I + \lambda\mathcal{L} + \mu\lambda K^{-1})^{-1/2}(I + \lambda\mathcal{L} + \mu\lambda K^{-1})^{-1/2} \\
&\qquad \cdots \times (\lambda\mu K^{-1})^{1/2}K^{-1/2}(I + \lambda\mathcal{L})^{-1}g_\rho.
\end{aligned}$$

As a consequence,

$$\|g_{\lambda,\mu} - g_\lambda\|_{L^2}^2 \le \lambda\mu \left\|K^{-1/2}(I + \lambda\mathcal{L})^{-1}g_\rho\right\|_{L^2}^2,$$

where we used the fact that $I + \lambda\mathcal{L} + \mu\lambda K^{-1} \succeq I$, so that $\left\|(I + \lambda\mathcal{L} + \mu\lambda K^{-1})^{-1/2}\right\|_{\mathrm{op}} \le 1$ (with $\|\cdot\|_{\mathrm{op}}$ the operator norm), and that

$$\begin{aligned}
\left\|(I + \lambda\mathcal{L} + \mu\lambda K^{-1})^{-1/2}(\lambda\mu K^{-1})^{1/2}\right\|_{\mathrm{op}}^2 &= \lambda\mu \left\|K^{-1/2}(I + \lambda\mathcal{L} + \mu\lambda K^{-1})^{-1}K^{-1/2}\right\|_{\mathrm{op}} \\
&= \lambda\mu \left\|(K + \lambda K^{1/2}\mathcal{L}K^{1/2} + \mu\lambda)^{-1}\right\|_{\mathrm{op}} \le 1.
\end{aligned}$$

We continue the proof with

$$\begin{aligned}
\left\|K^{-1/2}(I + \lambda\mathcal{L})^{-1}g_\rho\right\| &= \left\|\sum_{i=1}^r \frac{\lambda_i}{\lambda + \lambda_i} \langle g_\rho, e_i \rangle K^{-1/2}e_i\right\| \le \sum_{i=1}^r \frac{\lambda_i}{\lambda + \lambda_i} |\langle g_\rho, e_i \rangle| \left\|K^{-1/2}e_i\right\| \\
&\le \sum_{i=1}^r |\langle g_\rho, e_i \rangle| \left\|K^{-1/2}e_i\right\|_{L^2} \le \|g_\rho\|_{L^2} \left(\sum_{i\le r} \left\|K^{-1/2}e_i\right\|_{L^2}^2\right)^{1/2}.
\end{aligned}$$

Putting all the pieces together ends the proof. $\qquad\square$

### E.2.2 Vector concentration

We are left with the study of the variance $\|\hat{g}_p - g_{\lambda,\mu}\|$. To ease derivations, we denote $C = \Sigma + \lambda L$, $\hat{C} = \hat{\Sigma} + \lambda\hat{L}$, $\theta_\rho = S^\star g_\rho$, $\hat{\theta}_\rho = \widehat{S^\star g_\rho}$ and $P$ the projection from $\mathcal{H}$ to $\mathrm{Span}\{k_{X_i}\}_{i\le p}$. We have, for Tikhonov regularization

$$\begin{aligned}
\|\hat{g}_p - g_{\lambda,\mu}\|_{L^2} &= \left\|S\left(P(P\hat{C}P + \lambda\mu)^{-1}P\hat{\theta}_\rho - (C + \lambda\mu)^{-1}\theta_\rho\right)\right\|_{L^2} \\
&= \left\|\Sigma^{1/2}\left(P(P\hat{C}P + \lambda\mu)^{-1}P\hat{\theta}_\rho - (C + \lambda\mu)^{-1}\theta_\rho\right)\right\|_{\mathcal{H}}.
\end{aligned}$$

We begin by isolating the dependency to labelled data

$$\|\hat{g}_p - g_{\lambda,\mu}\|_{L^2} \leq \left\| \Sigma^{1/2} P (P\hat{C}P + \lambda\mu)^{-1} (\hat{\theta}_\rho - \theta_\rho) \right\|_{\mathcal{H}}$$
$$\cdots + \left\| \Sigma^{1/2} \left( P(P\hat{C}P + \lambda\mu)^{-1}\theta_\rho - (C + \lambda\mu)^{-1}\theta_\rho \right) \right\|_{\mathcal{H}}. \tag{24}$$

We will control the first term here, and the second term in the following subsection.

**Lemma 13** (Vector term). *When* $\left\| (C + \lambda\mu)^{-1/2} (\hat{C} - C)(C + \lambda\mu)^{-1/2} \right\|_{\mathrm{op}} \leq 1/2$, *we have*

$$\left\| \Sigma^{1/2} P (P\hat{C}P + \lambda\mu)^{-1} P(\hat{\theta}_\rho - \theta_\rho) \right\|_{\mathcal{H}} \leq 2 \left\| (C + \lambda\mu)^{-1/2}(\hat{\theta}_\rho - \theta_\rho) \right\|_{\mathcal{H}}. \tag{25}$$

*Proof.* We begin with the splitting

$$\left\| \Sigma^{1/2} P (P\hat{C}P + \lambda\mu)^{-1} P(\hat{\theta}_\rho - \theta_\rho) \right\|_{\mathcal{H}} \leq \left\| \Sigma^{1/2} P (P\hat{C}P + \lambda\mu)^{-1} P(C + \lambda\mu)^{1/2} \right\|_{\mathrm{op}}$$
$$\cdots \times \left\| (C + \lambda\mu)^{-1/2}(\hat{\theta}_\rho - \theta_\rho) \right\|_{\mathcal{H}}.$$

The first term will concentrate towards a matrix smaller than identity, while the second term concentrates towards zero. We can make those considerations more formal. Following basic properties with the Löwner order on operators, we have

$$(C + \lambda\mu)^{-1/2}(C - \hat{C})(C + \lambda\mu)^{-1/2} \preceq t$$
$$\Rightarrow \quad \hat{C} \succeq (1 - t)C - t\lambda\mu$$
$$\Rightarrow \quad P\hat{C}P \succeq (1 - t)PCP - t\lambda\mu P \succeq (1 - t)PCP - t\lambda\mu$$
$$\Rightarrow \quad P\hat{C}P + \lambda\mu \succeq (1 - t)(PCP + \lambda\mu)$$
$$\Rightarrow \quad (P\hat{C}P + \lambda\mu)^{-1} \preceq (1 - t)^{-1}(PCP + \lambda\mu)^{-1}$$
$$\Rightarrow \quad (C + \lambda\mu)^{1/2}P(P\hat{C}P + \lambda\mu)^{-1}P(C + \lambda\mu)^{1/2}$$
$$\preceq (1 - t)^{-1}(C + \lambda\mu)^{1/2}P(PCP + \lambda\mu)^{-1}P(C + \lambda\mu)^{1/2} \preceq (1 - t)^{-1},$$

where we have used the fact that the last operator is a projection. As a consequence, for any $t \in (0, 1)$, we have

$$\left\| (C + \lambda\mu)^{-1/2}(\hat{C} - C)(C + \lambda\mu)^{-1/2} \right\|_{\mathrm{op}} \leq t$$
$$\Rightarrow \quad \left\| (C + \lambda\mu)^{1/2}P(P\hat{C}P + \lambda\mu)^{-1}(C + \lambda\mu)^{1/2} \right\|_{\mathrm{op}} \leq (1 - t)^{-1}.$$
$$\Rightarrow \quad \left\| \Sigma^{1/2}P(P\hat{C}P + \lambda\mu)^{-1}P(C + \lambda\mu)^{1/2} \right\|_{\mathrm{op}} \leq (1 - t)^{-1}.$$

Where the last implication follows from the fact that $C + \lambda\mu = \Sigma + \lambda L + \lambda\mu \succeq \Sigma$. □

### E.2.3 Basis concentration

We are left with the study of the basis concentration with the number of unlabelled data.

**Lemma 14** (Basis term). *When* $\left\| (C + \lambda\mu)^{-1/2} (\hat{C} - C)(C + \lambda\mu)^{-1/2} \right\|_{\mathrm{op}} \leq 1/2$, *we have*

$$\left\| \Sigma^{1/2}(P(P\hat{C}P + \lambda\mu)^{-1} - (C + \lambda\mu)^{-1})\theta_\rho \right\|_{\mathcal{H}}$$
$$\leq 3 \left\| C^{1/2}(I - P) \right\|_{\mathrm{op}} \|g_\lambda\|_{\mathcal{H}} + 2 \left\| (C + \lambda\mu)^{-1/2}(\hat{C} - C)(C + \lambda\mu)^{-1}\theta_\rho \right\|_{\mathcal{H}}. \tag{26}$$

*Notice that Assumptions 1 and 2 imply* $\|g_\lambda\|_{\mathcal{H}} \leq c_a \|g_\rho\|_{L^2} < +\infty$.

*Proof.* First of all, using that $A^{-1} - B^{-1} = A^{-1}(B - A)B^{-1}$, notice that

$$\left\|\Sigma^{1/2}(P(P\hat{C}P + \lambda\mu)^{-1} - (C + \lambda\mu)^{-1})\theta_\rho\right\|_{\mathcal{H}}$$

$$= \left\|\Sigma^{1/2}P(P\hat{C}P + \lambda\mu)^{-1}P(C - \hat{C}P)(C + \lambda\mu)^{-1}\theta_\rho - \Sigma^{1/2}(I - P)(C + \lambda\mu)^{-1}\theta_\rho\right\|_{\mathcal{H}}$$

$$\leq \left\|\Sigma^{1/2}P(P\hat{C}P + \lambda\mu)^{-1}P(\hat{C}P - C)(C + \lambda\mu)^{-1}\theta_\rho\right\|_{\mathcal{H}} + \left\|\Sigma^{1/2}(I - P)(C + \lambda\mu)^{-1}\theta_\rho\right\|_{\mathcal{H}}$$

$$\leq \left\|\Sigma^{1/2}P(P\hat{C}P + \lambda\mu)^{-1}P(C + \lambda\mu)^{1/2}\right\|_{\mathrm{op}} \left\|(C + \lambda\mu)^{-1/2}P(\hat{C}P - C)(C + \lambda\mu)^{-1}\theta_\rho\right\|_{\mathcal{H}}$$

$$\cdots + \left\|\Sigma^{1/2}(I - P)\right\|_{\mathrm{op}} \left\|(C + \lambda\mu)^{-1}\theta_\rho\right\|_{\mathcal{H}}.$$

Because $\Sigma \preceq \Sigma + \lambda L = C$, we have $\left\|\Sigma^{1/2}(I - P)\right\|_{\mathrm{op}} \leq \left\|C^{1/2}(I - P)\right\|_{\mathrm{op}}$, and we also have

$$\left\|(C + \lambda\mu)^{-1}\theta_\rho\right\|_{\mathcal{H}} \leq \left\|C^{-1}\theta_\rho\right\|_{\mathcal{H}} = \left\|K^{-1/2}SC^{-1}\theta_\rho\right\|_{\mathcal{H}} = \left\|K^{-1/2}g_\lambda\right\|_{L^2} = \|g_\lambda\|_{\mathcal{H}}.$$

Recall, that, for any $t \in (0, 1)$, we have already shown that

$$\left\|(C + \lambda\mu)^{-1/2}(\hat{C} - C)(C + \lambda\mu)^{-1/2}\right\|_{\mathrm{op}} \leq t$$

$$\Rightarrow \quad \left\|\Sigma^{1/2}P(P\hat{C}P + \lambda\mu)^{-1}P(C + \lambda\mu)^{1/2}\right\|_{\mathrm{op}} \leq (1 - t)^{-1}.$$

We are left with one last term to work on

$$\left\|(C + \lambda\mu)^{-1/2}P(\hat{C}P - C)(C + \lambda\mu)^{-1}\theta_\rho\right\|_{\mathcal{H}} \leq \left\|(C + \lambda\mu)^{-1/2}P(\hat{C} - C)P(C + \lambda\mu)^{-1}\theta_\rho\right\|_{\mathcal{H}}$$

$$\cdots + \left\|(C + \lambda\mu)^{-1/2}C(I - P)(C + \lambda\mu)^{-1}\theta_\rho\right\|_{\mathcal{H}}.$$

We control the first term with the fact for $A, B, C$ three self-adjoint operators and $x$ a vector we have

$$\|APBPCx\| = \|APBPCx \otimes xCPBPA\|_{\mathrm{op}}^{1/2},$$

and that

$$PCx \otimes xCP \preceq Cx \otimes xC$$

$$\Rightarrow \quad PBPCx \otimes xCPBP \preceq BPCx \otimes xCPB \preceq BCx \otimes xCB$$

$$\Rightarrow \quad APBPCx \otimes xCPBPA \preceq ABCx \otimes xCBA,$$

so that

$$\left\|(C + \lambda\mu)^{-1/2}P(\hat{C} - C)P(C + \lambda\mu)^{-1}\theta_\rho\right\|_{\mathcal{H}} \leq \left\|(C + \lambda\mu)^{-1/2}(\hat{C} - C)(C + \lambda\mu)^{-1}\theta_\rho\right\|_{\mathcal{H}}.$$

We control the second term with

$$\left\|(C + \lambda\mu)^{-1/2}C(I - P)(C + \lambda\mu)^{-1}\theta_\rho\right\|$$

$$\leq \left\|(C + \lambda\mu)^{-1/2}C^{1/2}\right\| \left\|C^{1/2}(I - P)\right\| \left\|(C + \lambda\mu)^{-1}\theta_\rho\right\|.$$

Using that $(C + \lambda\mu)^{-1/2}C^{1/2} \preceq I$, we can add up everything to get the lemma.

For the part concerning $\|g_\lambda\|_{\mathcal{H}}$, notice that

$$\|g_\lambda\|_{\mathcal{H}} = \left\|K^{-1/2}g_\lambda\right\|_{L^2} = \left\|\sum_{i=1}^{r} \frac{\lambda_i}{\lambda_i + \lambda} \langle g_\rho, e_i\rangle K^{-1/2}e_i\right\|_{L^2} \leq \sum_{i=1}^{r} |g_\rho| \, e_i \left\|K^{-1/2}e_i\right\|$$

$$\leq \|g_\rho\|_{L^2} \left(\sum_{i=1}^{d} \left\|K^{-1/2}e_i\right\|_{L^2}^2\right)^{1/2} = c_a \|g_\rho\|_{L^2},$$

with $c_a$ defined as before. $\qquad\square$

### E.2.4 Conclusion

Based on the last subsections, we have proved the following proposition.

**Proposition 15** (Risk decomposition)**.** *Under the Assumptions 1 and 2, when* $\left\| (C + \lambda\mu)^{-1/2}(\hat{C} - C)(C + \lambda\mu)^{-1/2} \right\| \leq 1/2$,

$$
\begin{aligned}
\|\hat{g}_p - g_\rho\|_{L^2}^2 &\leq 4\lambda^2 \|\mathcal{L}g_\rho\|_{L^2}^2 + 4\lambda\mu c_a^2 \|g_\rho\|_{L^2}^2 + 8 \left\| (C + \lambda\mu)^{-1/2}(\hat{\theta}_\rho - \theta_\rho) \right\|_{\mathcal{H}}^2 \\
&\cdots + 12c_a^2 \left\| C^{1/2}(I - P) \right\|_{\text{op}}^2 \|g_\rho\|_{L^2}^2 + 8 \left\| (C + \lambda\mu)^{-1/2}(\hat{C} - C)(C + \lambda\mu)^{-1}\theta_\rho \right\|_{\mathcal{H}}^2 .
\end{aligned}
\tag{27}
$$

We are left with the quantification of the different convergences when the number of labelled and unlabelled data grows towards infinity. We will quantify those convergences based on concentration inequalities.

### E.3 Probabilistic inequalities

In this subsection, we bound each term appearing in Eq. (27) based on concentration inequalities.

#### E.3.1 Vector concentration

The concentration of $\hat{\theta}_\rho = \widehat{S^\star g_\rho}$ towards $\theta_\rho = S^\star g_\rho$ is controlled through Bernstein inequality.

**Theorem 2** (Concentration in Hilbert space [58])**.** *Let denote by $\mathcal{A}$ a Hilbert space and by $(\xi_i)$ a sequence of independent random vectors in $\mathcal{A}$ such that $\mathbb{E}[\xi_i] = 0$, that are bounded by a constant $M$, with finite variance $\sigma^2 = \mathbb{E}[\sum_{i=1}^n \|\xi_i\|^2]$. For any $t > 0$,*

$$
\mathbb{P}(\left\| \sum_{i=1}^n \xi_i \right\| \geq t) \leq 2\exp\left( -\frac{t^2}{2\sigma^2 + 2tM/3} \right).
$$

**Proposition 16** (Vector concentration)**.** *When $|Y|$ is bounded by a constant $c_{\mathcal{Y}}$, and $x \to k(x, x)$ by a constant $\kappa^2$, we have, with $\mathcal{D}_{n_\ell} \sim \rho^{\otimes n_\ell}$ the dataset generating the labelled data*

$$
\mathbb{P}_{\mathcal{D}_{n_\ell}} \left( \left\| (C + \lambda\mu)^{-1/2}(\hat{\theta}_\rho - \theta_\rho) \right\|_{\mathcal{H}} \geq t \right) \leq 2\exp\left( -\frac{n_\ell t^2}{2\sigma_\ell^2(\mu\lambda)^{-1} + 2tc_{\mathcal{Y}}(\mu\lambda)^{-1/2}\kappa/3} \right), \tag{28}
$$

*where $\sigma_\ell^2 \leq c_{\mathcal{Y}}^2 \operatorname{Tr}(\Sigma)$ is a variance parameter to relate with the variance of $Y(I + \lambda\mathcal{L})^{-1}\delta_X$ (where the randomness is inherited from $(X, Y) \sim \rho$).*

*Proof.* Recall that

$$
(C + \lambda\mu)^{-1/2}(\hat{\theta}_\rho - \theta_\rho) = (\Sigma + \lambda L + \lambda\mu)^{-1/2}(n_\ell^{-1} \sum_{i=1}^{n_\ell} Y_i k_{X_i} - \mathbb{E}_\rho[Y k_X])
$$

We want to apply Bernstein inequality to the vector $\xi_i = (\Sigma + \lambda L + \mu\lambda)^{-1/2} Y_i k_{X_i}$, after centering it. Let us denote by $c_{\mathcal{Y}}$ a bound on $|Y|$, $c_{\mathcal{Y}} \in \mathbb{R}$ exists since we have supposed $\rho$ of compact support. We have

$$
\begin{aligned}
\sigma^2 &= \mathbb{E}[\sum_{i=1}^{n_\ell} \|\xi_i - \mathbb{E}[\xi_i]\|^2] = n_\ell \mathbb{E}[\|\xi - \mathbb{E}[\xi_i]\|^2] \leq n_\ell \mathbb{E}[\|\xi\|^2] \\
&= n_\ell \mathbb{E}_{(X,Y)\sim\rho} \left[ Y^2 \left\langle k_X, (\Sigma + \lambda L + \mu\lambda)^{-1}k_X \right\rangle \right] \\
&\leq n_\ell c_{\mathcal{Y}}^2 \mathbb{E}_{X\sim\rho_{\mathcal{X}}} \left[ \left\langle k_X, (\Sigma + \lambda L + \mu\lambda)^{-1}k_X \right\rangle \right] \\
&= n_\ell c_{\mathcal{Y}}^2 \operatorname{Tr}\left( (\Sigma + \lambda L + \mu\lambda)^{-1}\Sigma \right) \\
&\leq n_\ell c_{\mathcal{Y}}^2 \operatorname{Tr}(\Sigma) \left\| (\Sigma + \lambda L + \mu\lambda)^{-1} \right\|_{\text{op}} \leq n_\ell c_{\mathcal{Y}}^2 \operatorname{Tr}(\Sigma)(\mu\lambda)^{-1}.
\end{aligned}
$$

Note that we have proceed with a generic upper bound, but we expect this variance, which is related to the variance of $Y(I + \lambda\mathcal{L} + \lambda\mu K^{-1})^{-1}\delta_X$ to be potentially much smaller – if we remove the term in $P$ the vector concentration is the concentration of the vector $S(S^*S + \lambda S^\star \mathcal{L}S + \lambda\mu)^{-1}Y k_X \simeq$

$K^{1/2}(K + \lambda K^{1/2}\mathcal{L}K^{1/2} + \lambda\mu)^{-1}K^{1/2}S^{-\star}Yk_X = (I + \lambda L + \lambda\mu K^{-1})^{-1}YS^{-\star}k_X \simeq (I + \lambda L + \lambda\mu K^{-1})^{-1}Y\delta_X$. To capture this fact, we will write $\sigma^2 \leq n_\ell\sigma_\ell^2(\mu\lambda)^{-1}$, with $\sigma_\ell = c_{\mathcal{Y}}\operatorname{Tr}(\Sigma)^{1/2}$ in our analysis, but potentially much smaller under refined hypothesis and in practice. Similarly to the bound on the variance, we have

$$\|\xi - \mathbb{E}[\xi]\| \leq \|\xi\| = \left\|(\Sigma + \lambda L + \mu\lambda)^{-1/2}Y_ik_{X_i}\right\| \leq (\mu\lambda)^{-1/2}c_{\mathcal{Y}}\kappa,$$

with $\kappa$ an upper bound on $k(x,x)^{1/2}$ for $x \in \operatorname{supp}\rho_{\mathcal{X}}$. As a consequence, applying Bernstein concentration inequality, we get, for any $t > 0$,

$$\mathbb{P}_{\mathcal{D}_{n_\ell}}\left(\left\|n_\ell^{-1}\sum_{i=1}^{n_\ell}\xi_i - E[\xi_i]\right\| \geq t\right) \leq 2\exp\left(-\frac{n_\ell t^2}{2\sigma_\ell^2(\mu\lambda)^{-1} + 2tc_{\mathcal{Y}}(\mu\lambda)^{-1/2}\kappa/3}\right).$$

This ends the proof. $\qquad\square$

### E.3.2 Operator concentration

The convergence of $\hat{C}$ towards $C$ is controlled with Bernstein inequality for self-adjoint operators.

**Theorem 3** (Bernstein inequality for self-adjoint [36])**.** *Let $\mathcal{A}$ be a separable Hilbert space, and $(\xi_i)$ a sequence of independent random self-adjoint operators operators on $\mathcal{A}$ Assume that $(\xi_i)$ are bounded by $M \in \mathbb{R}$, in the sense that, almost everywhere, $\|\xi\|_{\mathrm{op}} < M$, and have a finite variance $\sigma^2 = \left\|\sum_{i=1}^n \mathbb{E}[\xi_i^2]\right\|_{\mathrm{op}}$. For any $t > 0$,*

$$\mathbb{P}\left(\left\|\sum_{i=1}^n(\xi_i - \mathbb{E}[\xi_i])\right\|_{\mathrm{op}} > t\right)$$

$$\leq 2\left(1 + 6\frac{\sigma^2 + Mt/3}{t^2}\right)\frac{\operatorname{Tr}\left(\sum_{i=1}^n \mathbb{E}[\xi_i^2]\right)}{\left\|\sum_{i=1}^n \mathbb{E}[\xi_i^2]\right\|_{\mathrm{op}}}\exp\left(-\frac{t^2}{2\sigma^2 + 2tM/3}\right).$$

**Proposition 17** (Operator concentration)**.** *When $x \to k(x,x)$ is bounded by $\kappa^2$ a nd $x \to \partial_{1,j}\partial_{2,j}k(x,x)$ is bounded by $\kappa_j^2$, we have*

$$\mathbb{P}_{\mathcal{D}_n}\left(\left\|(C + \mu\lambda)^{-\frac{1}{2}}(C - \hat{C})(C + \mu\lambda)^{-\frac{1}{2}}\right\|_{\mathrm{op}} > 1/2\right) \leq \left(2 + 56\frac{\kappa^2 + \lambda\sum_{i=1}^d \kappa_j^2}{\lambda\mu n}\right)$$

$$\cdots \times (1 + \lambda\mu\|C\|_{\mathrm{op}}^{-1})\frac{\kappa^2 + \lambda\sum_{j=1}^d \kappa_j^2}{\lambda\mu}\exp\left(-\frac{\lambda\mu n}{10\left(\kappa^2 + \lambda\sum_{j=1}^d \kappa_j^2\right)}\right). \tag{29}$$

*Proof.* We want to apply the precedent concentration inequality to

$$\xi_i = (\Sigma + \lambda L + \lambda\mu)^{-1/2}(k_{X_i}\otimes k_{X_i} + \lambda\sum_{j=1}^d \partial_j k_{X_i}\otimes\partial_j k_{X_i})(\Sigma + \lambda L + \lambda\mu)^{-1/2},$$

since we have, based on the fact that $C = \Sigma + \lambda L$ and that $\Sigma = \mathbb{E}[k_X\otimes k_X]$ and $L = \mathbb{E}[\sum_{j=1}^n \partial_j k_X\otimes\partial_j k_X]$,

$$\left\|(C + \lambda\mu)^{-1/2}(\hat{C} - C)(C + \lambda\mu)^{-1/2}\right\|_{\mathrm{op}} = n^{-1}\left\|\sum_{i=1}^n \xi_i - \mathbb{E}[\xi_i]\right\|_{\mathrm{op}}.$$

We bound $\xi$ with

$$
\begin{aligned}
\|\xi\|_{\mathrm{op}} &= \left\| (C+\mu\lambda)^{-\frac{1}{2}} \left( k_X \otimes k_X + \lambda \sum_{j=1}^{d} \partial_j k_X \otimes \partial_j k_X \right) (C+\mu\lambda)^{-\frac{1}{2}} \right\|_{\mathrm{op}} \\
&\leq \mathrm{Tr}\left( (C+\mu\lambda)^{-\frac{1}{2}} \left( k_X \otimes k_X + \lambda \sum_{j=1}^{d} \partial_j k_X \otimes \partial_j k_X \right) (C+\mu\lambda)^{-\frac{1}{2}} \right) \\
&= \mathrm{Tr}\left( (C+\mu\lambda)^{-\frac{1}{2}} k_X \otimes k_X (C+\mu\lambda)^{-\frac{1}{2}} \right) \\
&\quad \cdots + \lambda \sum_{j=1}^{d} \mathrm{Tr}\left( (C+\mu\lambda)^{-\frac{1}{2}} \partial_j k_X \otimes \partial_j k_X (C+\mu\lambda)^{-\frac{1}{2}} \right) \\
&= \left\| (C+\mu\lambda)^{-\frac{1}{2}} k_X \right\|_{\mathcal{H}}^2 + \lambda \sum_{j=1}^{d} \left\| (C+\mu\lambda)^{-\frac{1}{2}} \partial_j k_X \right\|_{\mathcal{H}}^2 \leq (\lambda\mu)^{-1}\left( \kappa^2 + \lambda \sum_{j=1}^{d} \kappa_j^2 \right).
\end{aligned}
$$

With $\kappa^2$ an upper bound on the kernel $k$ and $\kappa_j^2$ an upper bound on $\partial_{1,j}\partial_{2,j}k$. For the variance we have, using Löwner order,

$$
\begin{aligned}
\mathbb{E}[\xi^2] &\preceq \sup_{X\in\mathcal{X}} \|\xi(X)\|_{\mathrm{op}} \mathbb{E}[\xi] \preceq (\lambda\mu)^{-1}\left( \kappa^2 + \lambda \sum_{j=1}^{d} \kappa_j^2 \right) \mathbb{E}[\xi] \\
&= (\lambda\mu)^{-1}\left( \kappa^2 + \lambda \sum_{j=1}^{d} \kappa_j^2 \right)(C+\lambda)^{-1}C \preceq (\lambda\mu)^{-1}\left( \kappa^2 + \lambda \sum_{j=1}^{d} \kappa_j^2 \right).
\end{aligned}
$$

Therefore, we get for any $t > 0$,

$$
\begin{aligned}
&\mathbb{P}_{\mathcal{D}_n}\left( \left\| (C+\mu\lambda)^{-\frac{1}{2}}(C-\hat{C})(C+\mu\lambda)^{-\frac{1}{2}} \right\|_{\mathrm{op}} > t \right) \\
&\leq 2\left( 1 + 6\frac{(\kappa^2 + \lambda\sum_{i=1}^{d}\kappa_j^2)(1+t/3)}{\lambda\mu n t^2} \right) \frac{\|C\|_{\mathrm{op}} + \lambda\mu}{\|C\|_{\mathrm{op}}} \mathrm{Tr}\left( (C+\lambda)^{-1}C \right) \\
&\quad \cdots \times \exp\left( -\frac{n t^2}{2(\lambda\mu)^{-1}\left( \kappa^2 + \lambda\sum_{j=1}^{d}\kappa_j^2 \right)(1+t/3)} \right).
\end{aligned}
$$

Remark that

$$
\mathrm{Tr}\left( (C+\mu\lambda)^{-1}C \right) \leq \left\| (C+\mu\lambda)^{-1} \right\|_{\mathrm{op}} \mathrm{Tr}(C) \leq (\lambda\mu)^{-1}(\kappa^2 + \lambda\sum_{j=1}^{d}\kappa_j^2).
$$

Taking $t = 1/2$ ends the lemma. □

### E.3.3 Basis concentration

Similarly we could control $\left\| (C+\lambda\mu)^{-1/2}(\hat{C}-C)(C+\lambda\mu)^{-1}\theta_\rho \right\|_{\mathcal{H}}$ by using concentration of self-adjoint, yet this will lead to laxer bounds, than using concentration on vectors.

**Proposition 18** (Basis concentration). *When $x \to k(x,x)$ is bounded by $\kappa^2$, $x \to \partial_{1,j}\partial_{2,j}k(x,x)$ is bounded by $\kappa_j^2$, with Assumptions 1 and 2, we have*

$$
\mathbb{P}_{\mathcal{D}_n}\left( \left\| (C+\mu\lambda)^{-\frac{1}{2}}(C-\hat{C})(C+\mu\lambda)^{-1}\theta_\rho \right\|_{\mathcal{H}} > t \right) \leq 2\exp\left( -\frac{\mu\lambda n t^2}{2c_1(c_1 + \lambda^{1/2}\mu^{1/2}t/3)} \right). \tag{30}
$$

*with $c_1 = (\kappa^2 + \lambda\sum_{i=1}^{d}\kappa_j^2)c_a \|g_\rho\|_{L^2}$.*

*Proof.* We want to apply Bernstein concentration inequality to the vectors

$$\xi_i = (C + \mu\lambda)^{-1/2} \left( k_{X_i} \otimes k_{X_i} + \lambda \sum_{j=1}^{d} \partial_j k_{X_i} \otimes \partial_j k_{X_i} \right) (C + \lambda\mu)^{-1}\theta_\rho,$$

since

$$\left\| (C + \mu\lambda)^{-\frac{1}{2}}(C - \hat{C})(C + \mu\lambda)^{-1}\theta_\rho \right\|_{\mathcal{H}} = n^{-1} \left\| \sum_{i=1}^{n} \xi_i - \mathbb{E}[\xi_i] \right\|_{\mathcal{H}}.$$

We bound $\xi$, reusing prior derivations, with

$$\|\xi_i\|_{\mathcal{H}} = \left\| (C + \mu\lambda)^{-1/2} \left( k_{X_i} \otimes k_{X_i} + \lambda \sum_{j=1}^{d} \partial_j k_{X_i} \otimes \partial_j k_{X_i} \right) (C + \lambda\mu)^{-1}\theta_\rho \right\|_{\mathcal{H}}$$

$$\leq \left\| (C + \mu\lambda)^{-1/2} \right\|_{\text{op}} \left\| \left( k_{X_i} \otimes k_{X_i} + \lambda \sum_{j=1}^{d} \partial_j k_{X_i} \otimes \partial_j k_{X_i} \right) \right\|_{\text{op}} \left\| (C + \mu\lambda)^{-1}\theta_\rho \right\|_{\mathcal{H}}.$$

$$\leq (\mu\lambda)^{-1/2}(\kappa^2 + \lambda \sum_{i=1}^{d} \kappa_j^2) c_a \|g_\rho\|_{L^2} .$$

For the variance, we have, similarly to prior derivations,

$$\mathbb{E}[\|\xi\|^2] \leq \sup_{X \in \mathcal{X}} \left\| k_X \otimes k_X + \lambda \sum_{j=1}^{d} \partial_j k_X \otimes \partial_j k_X \right\|_{\text{op}} \left\| (C + \lambda\mu)^{-1}\theta_\rho \right\|^2$$

$$\cdots \times \mathbb{E} \left[ \left\| (C + \mu\lambda)^{-1} \left( k_X \otimes k_X + \lambda \sum_{j=1}^{d} \partial_j k_{X_i} \otimes \partial_j k_X \right) \right\|_{\text{op}} \right]$$

$$\leq \left( \kappa^2 + \lambda \sum_{i=1}^{d} \kappa_i^2 \right) c_a^2 \|g_\rho\|_{L^2}^2$$

$$\cdots \mathbb{E} \left[ \left\| (C + \mu\lambda)^{-1} k_X \otimes k_X \right\|_{\text{op}} + \lambda \sum_{j=1}^{d} \left\| (C + \mu\lambda)^{-1} \partial_j k_{X_i} \otimes \partial_j k_X \right\|_{\text{op}} \right]$$

$$= \left( \kappa^2 + \lambda \sum_{i=1}^{d} \kappa_i^2 \right) c_a^2 \|g_\rho\|_{L^2}^2 \operatorname{Tr} \left( (C + \mu\lambda)^{-1} C \right)$$

$$\leq (\lambda\mu)^{-1} \left( \kappa^2 + \lambda \sum_{i=1}^{d} \kappa_i^2 \right)^2 c_a^2 \|g_\rho\|_{L^2}^2 .$$

As a consequence, using Bernstein inequality,

$$\mathbb{P} \left( n^{-1} \left\| \sum_{i=1}^{n} \xi_i - \mathbb{E}[\xi_i] \right\| > t \right) \leq 2 \exp \left( -\frac{\mu\lambda n t^2}{2c_1(c_1 + \lambda^{1/2}\mu^{1/2}t/3)}, \right)$$

with $c_1 = (\kappa^2 + \lambda \sum_{i=1}^{d} \kappa_j^2) c_a \|g_\rho\|_{L^2}$. Note that we have bound naively the variable $\xi$ and its variance, but we have shown how appears $\sup_{X \in \mathcal{X}} \left\| (C + \lambda\mu)^{-1} k_X \right\| + \lambda \sum_{i=1}^{d} \left\| (C + \lambda\mu)^{-1} \partial_i k_X \right\|$ and $\operatorname{Tr}((C + \lambda\mu)^{-1} C)$, which under interpolation and capacity assumptions but be controlled in a better fashion. $\square$

### E.3.4 Low-rank approximation

We now switch to Nyström approximation.

**Proposition 19** (Low-rank approximation). *When $x \to k(x, x)$ is bounded by $\kappa^2$, for any $p \in \mathbb{N}$ and $t > 0$, we have*

$$\mathbb{P}_{\mathcal{D}_p} \left( \left\| (I - P)\Sigma^{1/2} \right\|^2 > t \right) \leq \left( 2 + \frac{116\kappa^2}{tp} \right) (2 + t \left\| \Sigma \right\|_{\mathrm{op}}^{-1}) \frac{\kappa^2}{t} \exp \left( -\frac{pt}{10\kappa^2} \right),$$

*Proof.* Reusing Proposition 3 of Rudi et al. [45], for any $\gamma > 0$, we have, with $P$ the projection on Span $\{k_{X_i}\}_{i \leq p}$ and $\hat{\Sigma} = p^{-1} \sum_{i=1}^{p} k_{X_i} \otimes k_{X_i}$,

$$\left\| (I - P)\Sigma^{1/2} \right\|^2 \leq \gamma \left\| (\hat{\Sigma} + \gamma)^{-1/2} \Sigma^{1/2} \right\|_{\mathrm{op}}^2 \leq \gamma \left\| \Sigma^{1/2} (\hat{\Sigma} + \gamma)^{-1} \Sigma^{1/2} \right\|_{\mathrm{op}}.$$

As a consequence, skipping derivations that can be retaken from our precedent proofs,

$$\mathbb{P}_{\mathcal{D}_p} \left( \left\| (I - P)\Sigma^{1/2} \right\|^2 > t \right) \leq \inf_{\gamma > 0} \mathbb{P}_{\mathcal{D}_p} \left( \gamma \left\| \Sigma^{1/2} (\hat{\Sigma} + \gamma)^{-1} \Sigma^{1/2} \right\|_{\mathrm{op}} > t \right)$$

$$\leq \inf_{\gamma > 0} \mathbb{P}_{\mathcal{D}_p} \left( \left\| (\Sigma + \gamma)^{-1/2} (\hat{\Sigma} - \Sigma)(\Sigma + \gamma)^{-1/2} \right\|_{\mathrm{op}} > (1 - \gamma t^{-1}) \right)$$

$$\leq \inf_{\gamma > 0} \left( 2 + 56 \frac{\kappa^2}{\gamma p} \right) (1 + \gamma \left\| \Sigma \right\|_{\mathrm{op}}^{-1}) \frac{\kappa^2}{\gamma} \exp \left( -\frac{p\gamma u^2}{2\kappa^2 (1 + u/3)} \right),$$

with $u = (1 - \gamma t^{-1})$. Taking $\gamma = t/2$, this term is simplified as

$$\mathbb{P}_{\mathcal{D}_p} \left( \left\| (I - P)\Sigma^{1/2} \right\|^2 > t \right) \leq \left( 2 + 116 \frac{\kappa^2}{tp} \right) (2 + t \left\| \Sigma \right\|_{\mathrm{op}}^{-1}) \frac{\kappa^2}{t} \exp \left( -\frac{pt}{10\kappa^2} \right),$$

which is the object of this proposition. $\qquad \square$

**Lemma 20.** *When $L \leq c_d \Sigma^a$, we have*

$$\left\| (I - P)C^{1/2} \right\|_{\mathrm{op}}^2 \leq \left\| (I - P)\Sigma^{1/2} \right\|_{\mathrm{op}}^2 + c_d \lambda \left\| (I - P)\Sigma^{1/2} \right\|_{\mathrm{op}}^{2a}. \tag{31}$$

*Proof.* This follows from the fact that

$$\left\| C^{1/2}(I - P) \right\|_{\mathrm{op}}^2 = \left\| (I - P)C(I - P) \right\|_{\mathrm{op}} = \left\| (I - P)(\Sigma + \lambda L)(I - P) \right\|_{\mathrm{op}}$$

$$\leq \left\| (I - P)\Sigma(I - P) \right\|_{\mathrm{op}} + \lambda \left\| (I - P)L(I - P) \right\|_{\mathrm{op}}$$

$$\leq \left\| (I - P)\Sigma(I - P) \right\|_{\mathrm{op}} + \lambda c_d \left\| (I - P)\Sigma^a (I - P) \right\|_{\mathrm{op}}$$

$$= \left\| (I - P)\Sigma^{1/2} \right\|_{\mathrm{op}}^2 + \lambda c_d \left\| (I - P)\Sigma^{a/2} \right\|_{\mathrm{op}}^2$$

$$= \left\| (I - P)\Sigma^{1/2} \right\|_{\mathrm{op}}^2 + \lambda c_d \left\| (I - P)^a \Sigma^{a/2} \right\|_{\mathrm{op}}^2$$

$$\leq \left\| (I - P)\Sigma^{1/2} \right\|_{\mathrm{op}}^2 + \lambda c_d \left\| (I - P)\Sigma^{1/2} \right\|_{\mathrm{op}}^{2a},$$

where we used the fact that $(I - P)^a = (I - P)$ and that $\|A^s B^s\| \leq \|AB\|^s$ for $s \in [0, 1]$ and $A, B$ positive self-adjoint. $\qquad \square$

### E.4 Averaged excess of risk - Ending the proof

Based on the precedent excess of risk decomposition, and precedent concentration inequalities, we have all the elements to derive convergence rates of our algorithm. We will enunciate this convergence in term of the averaged excess of risk of $\mathbb{E}_{\mathcal{D}_n} \left\| \|\hat{g}_p - g_\rho\|_{L^2}^2 \right\|$.

**Lemma 21.** *Under Assumptions 1 and 2,*

$$\mathbb{E}_{\mathcal{D}_n}\left[\|\hat{g}_p - g_\rho\|_{L^2}^2\right] \leq 4c_{\mathcal{Y}}^2 \, \mathbb{P}\left(\left\|(C+\lambda\mu)^{-1/2}(\hat{C}-C)(C+\lambda\mu)^{-1/2}\right\| \leq 1/2\right)$$

$$\cdots + 4\lambda^2 \|\mathcal{L}g_\rho\|_{L^2}^2 + 4\lambda\mu c_a^2 \|g_\rho\|_{L^2}^2$$

$$\cdots + 8\,\mathbb{E}_{\mathcal{D}_n}\left[\left\|(C+\lambda\mu)^{-1/2}(\hat{\theta}_\rho - \theta_\rho)\right\|_{\mathcal{H}}^2\right] + 12c_a^2 \|g_\rho\|_{L^2}^2 \, \mathbb{E}_{\mathcal{D}_n}\left[\left\|C^{1/2}(I-P)\right\|_{\text{op}}^2\right] \quad (32)$$

$$\cdots + 8\,\mathbb{E}_{\mathcal{D}_n}\left[\left\|(C+\lambda\mu)^{-1/2}(\hat{C}-C)(C+\lambda\mu)^{-1}\theta_\rho\right\|_{\mathcal{H}}^2\right].$$

*Proof.* We proceed using the fact that $\mathbb{E}[X] = \mathbb{E}[X \mid {}^cA]\,\mathbb{P}({}^cA) + \mathbb{E}[X \mid A]\,\mathbb{P}(A) \leq \sup X \, \mathbb{P}({}^cA) + \mathbb{E}[X \mid A]\,\mathbb{P}(A)$, with $A = \left\{\mathcal{D}_n \,\middle|\, \left\|(C+\lambda\mu)^{-1/2}(\hat{C}-C)(C+\lambda\mu)^{-1/2}\right\| \leq 1/2\right\}$,

$$\mathbb{E}_{\mathcal{D}_n}\left[\|\hat{g}_p - g_\rho\|_{L^2}^2\right] \leq \sup_{\mathcal{D}_n}\|\hat{g}_p - g_\rho\|^2 \, \mathbb{P}\left({}^cA\right) + \mathbb{E}_{\mathcal{D}_n}\left[\|\hat{g}_p - g_\rho\|^2 \,\middle|\, A\right]\mathbb{P}(A).$$

When $Y$ is bounded by $c_{\mathcal{Y}}$, because $g_\rho$ is a convex combination of $Y$, we know that $\|g_\rho\|_{L^2} \leq c_{\mathcal{Y}}$, as a consequence, we can clip $\hat{g}_p$ to $[-c_{\mathcal{Y}}, c_{\mathcal{Y}}]$, which will only improve the estimation of $g_\rho$, as a consequence, we can consider the clipping estimate for which we have $\sup_{\mathcal{D}_n}\|\hat{g}_p - g_\rho\|_{L^2}^2 \leq 4c_{\mathcal{Y}}^2$. Regarding the second part, we have already decomposed the risk under the event $A = \left\{\mathcal{D}_n \,\middle|\, \left\|(C+\lambda\mu)^{-1/2}(\hat{C}-C)(C+\lambda\mu)^{-1/2}\right\| \leq 1/2\right\}$. As a consequence, we have

$$\mathbb{E}_{\mathcal{D}_n}\left[\|\hat{g}_p - g_\rho\|_{L^2}^2\right] \leq 4c_{\mathcal{Y}}^2 \, \mathbb{P}({}^cA) + 4\lambda^2 \|\mathcal{L}g_\rho\|_{L^2}^2 \, \mathbb{P}(A) + 4\lambda\mu c_a^2 \|g_\rho\|_{L^2}^2 \, \mathbb{P}(A)$$

$$\cdots + 8\,\mathbb{E}_{\mathcal{D}_n}\left[\left\|(C+\lambda\mu)^{-1/2}(\hat{\theta}_\rho - \theta_\rho)\right\|_{\mathcal{H}}^2 \,\middle|\, A\right]\mathbb{P}(A)$$

$$\cdots + 12c_a^2 \|g_\rho\|_{L^2}^2 \, \mathbb{E}_{\mathcal{D}_n}\left[\left\|C^{1/2}(I-P)\right\|_{\text{op}}^2 \,\middle|\, A\right]\mathbb{P}(A)$$

$$\cdots + 8\,\mathbb{E}_{\mathcal{D}_n}\left[\left\|(C+\lambda\mu)^{-1/2}(\hat{C}-C)(C+\lambda\mu)^{-1}\theta_\rho\right\|_{\mathcal{H}}^2 \,\middle|\, A\right]\mathbb{P}(A).$$

To control the conditional expectation, we use that, when $X$ is positive

$$\mathbb{E}\left[X \mid A\right]P(A) = \mathbb{E}[X] - \mathbb{E}\left[X \mid {}^cA\right]\mathbb{P}({}^cA) \leq \mathbb{E}[X].$$

This ends the proof. $\qquad\square$

Based on deviation inequalities, we can control expectations based on the equality, for $X$ positive, $\mathbb{E}[X] = \int_0^{+\infty} \mathbb{P}(X > t)\, \mathrm{d}t$.

**Lemma 22.** *In the setting of the paper,*

$$\mathbb{E}_{\mathcal{D}_n}\left[\left\|(C+\lambda\mu)^{-1/2}(\hat{\theta}_\rho - \theta_\rho)\right\|_{\mathcal{H}}^2\right] \leq 8\sigma_\ell^2 (n_\ell\mu\lambda)^{-1} + 8c_{\mathcal{Y}}^2\kappa^2(n_\ell^2\mu\lambda)^{-1}. \quad (33)$$

*Proof.* First, recall that

$$\mathbb{P}\left(\left\|(C+\lambda\mu)^{-1/2}(\hat{\theta}_\rho - \theta_\rho)\right\|_{\mathcal{H}} > t\right) \leq 2\exp\left(-\frac{n_\ell t^2}{2\sigma_\ell^2(\mu\lambda)^{-1} + 2tc_{\mathcal{Y}}(\lambda\mu)^{-1/2}\kappa/3}\right)$$

$$\leq 2\exp\left(-\frac{n_\ell t^2}{2\max\left(2\sigma_\ell^2(\mu\lambda)^{-1}, 2tc_{\mathcal{Y}}(\lambda\mu)^{-1/2}\kappa/3\right)}\right)$$

$$\leq 2\exp\left(-\frac{n_\ell\mu\lambda t^2}{4\sigma_\ell^2}\right) + 2\exp\left(-\frac{3n_\ell\mu^{1/2}\lambda^{1/2}t}{4c_{\mathcal{Y}}\kappa}\right).$$

As a consequence

$$\mathbb{E}\left[\left\|(C+\lambda\mu)^{-1/2}(\hat{\theta}_\rho - \theta_\rho)\right\|_{\mathcal{H}}^2\right] = \int_0^{+\infty} \mathbb{P}\left(\left\|(C+\lambda\mu)^{-1/2}(\hat{\theta}_\rho - \theta_\rho)\right\|_{\mathcal{H}}^2 > t\right) dt$$

$$\leq 2\int \exp\left(-\frac{n_\ell\mu\lambda t}{4\sigma_\ell^2}\right) dt + 2\int \exp\left(-\frac{3n_\ell\mu^{1/2}\lambda^{1/2}t^{1/2}}{4c_{\mathcal{Y}}\kappa}\right) dt.$$

$$= 8\sigma_\ell^2(n_\ell\mu\lambda)^{-1} + \frac{64c_{\mathcal{Y}}^2\kappa^2}{9}(n_\ell^2\mu\lambda)^{-1}.$$

This is the result stated in the lemma. $\qquad\square$

**Lemma 23.** *In the setting of the paper,*

$$\mathbb{E}_{\mathcal{D}_n}\left[\left\|(C+\lambda\mu)^{-1/2}(\hat{C}-C)(C+\lambda\mu)^{-1}\theta_\rho\right\|_{\mathcal{H}}^2\right] \leq 8(\kappa^2+\lambda\partial\kappa^2)^2c_a^2\left\|g_\rho\right\|_{L^2}^2 \tag{34}$$
$$\cdots \times \left((\mu\lambda n)^{-1} + (\mu\lambda n^2)^{-1}\right),$$

*with $\partial\kappa^2 = \sum_{i=1}^d \kappa_i^2$.*

*Proof.* Let us denote by $A$ the quantity $\left\|(C+\lambda\mu)^{-1/2}(\hat{C}-C)(C+\lambda\mu)^{-1}\theta_\rho\right\|_{\mathcal{H}}$, and $\partial\kappa^2 = \sum_{i=1}^d \kappa_j^2$. Recall that

$$\mathbb{P}(A > t) \leq 2\exp\left(-\frac{\mu\lambda nt^2}{2c_1(c_1+\lambda^{1/2}\mu^{1/2}t/3)}\right)$$

$$\leq 2\exp\left(-\frac{\mu\lambda nt^2}{4c_1^2}\right) + 2\exp\left(-\frac{3(\mu\lambda)^{1/2}nt}{4c_1}\right).$$

We conclude the proof similarly to the precedent lemma. $\qquad\square$

**Lemma 24.** *Under Assumption 3,*

$$\mathbb{E}_{\mathcal{D}_n}\left[\left\|C^{1/2}(I-P)\right\|_{\mathrm{op}}^2\right] \leq \left(\frac{10\kappa^2\log(p)}{p} + \frac{10^a\kappa^{2a}c_d\lambda\log(p)^a}{p^a}\right)$$
$$\cdots \times \left(1 + \frac{2\kappa^2}{\left\|\Sigma\right\|_{\mathrm{op}}\log(p)}\left(1 + \frac{6}{\log(p)}\right)\left(\frac{1}{p} + \frac{1}{5\log(p)}\right)\right). \tag{35}$$

*Proof.* Once again, this result comes from integration of the tail bound obtained on $\left\|C^{1/2}(I-P)\right\|_{\mathrm{op}}^2$ through the one we have on $\left\|\Sigma^{1/2}(I-P)\right\|_{\mathrm{op}}^2$ and the fact that $\left\|C^{1/2}(I-P)\right\|_{\mathrm{op}}^2 \leq \left\|\Sigma^{1/2}(I-P)\right\|_{\mathrm{op}}^2 + c_d\lambda\left\|\Sigma^{1/2}(I-P)\right\|_{\mathrm{op}}^{2a}$. For any $a, b > 0$, we have

$$\mathbb{E}_{\mathcal{D}_n}\left[\left\|\Sigma^{1/2}(I-P)\right\|_{\mathrm{op}}^2\right] = \int_0^\infty \mathbb{P}_{\mathcal{D}_n}\left(\left\|\Sigma^{1/2}(I-P)\right\|_{\mathrm{op}}^2 > t\right) dt$$

$$\leq \int_0^\infty \min\left\{1, 2\kappa^2\left\|\Sigma\right\|_{\mathrm{op}}^{-1}\left(1 + \frac{58\kappa^2}{tp}\right)\left(1 + \frac{2\kappa^2}{t}\right)\exp\left(-\frac{pt}{10\kappa^2}\right)\right\} dt$$

$$= \frac{10\kappa^2 a}{p}\int_0^\infty \min\left\{1, 2\kappa^2\left\|\Sigma\right\|_{\mathrm{op}}^{-1}\left(1 + \frac{58}{10au}\right)\left(1 + \frac{p}{5au}\right)\exp\left(-au\right)\right\} du$$

$$\leq \frac{10\kappa^2 a}{p}\left(b + \int_b^\infty 2\kappa^2\left\|\Sigma\right\|_{\mathrm{op}}^{-1}\left(1 + \frac{6}{au}\right)\left(1 + \frac{p}{5au}\right)\exp\left(-au\right) du\right)$$

$$\leq \frac{10\kappa^2}{p}\left(ab + 2\kappa^2\left\|\Sigma\right\|_{\mathrm{op}}^{-1}\left(1 + \frac{6}{ab}\right)\left(1 + \frac{p}{5ab}\right)\exp\left(-ab\right)\right).$$

This last quantity is optimized for $ab = \log(p)$, which leads to the first part of lemma. Similarly

$$
\mathbb{E}_{\mathcal{D}_n}\left[\left\|\Sigma^{1/2}(I-P)\right\|_{\mathrm{op}}^{2a}\right] = \int_0^\infty \mathbb{P}_{\mathcal{D}_n}\left(\left\|\Sigma^{1/2}(I-P)\right\|_{\mathrm{op}}^{2a} > t\right)\mathrm{d}t
$$

$$
= \int_0^\infty \mathbb{P}_{\mathcal{D}_n}\left(\left\|\Sigma^{1/2}(I-P)\right\|_{\mathrm{op}}^2 > t^{1/a}\right)\mathrm{d}t
$$

$$
\leq \int_0^\infty \min\left\{1, 2\kappa^2\left\|\Sigma\right\|_{\mathrm{op}}^{-1}\left(1+\frac{58\kappa^2}{t^{1/a}p}\right)\left(1+\frac{2\kappa^2}{t^{1/a}}\right)\exp\left(-\frac{pt^{1/a}}{10\kappa^2}\right)\right\}\mathrm{d}t
$$

$$
= \frac{10^a\kappa^{2a}ac^a}{p^a}\int_0^\infty \min\left\{u^{a-1}, 2\kappa^2\left\|\Sigma\right\|_{\mathrm{op}}^{-1}\left(1+\frac{58}{10cu}\right)\left(1+\frac{p}{5cu}\right)\frac{1}{u^{1-a}}\exp\left(-cu\right)\right\}\mathrm{d}u
$$

$$
\leq \frac{10^a\kappa^{2a}ac^a}{p^a}\left(\frac{b^a}{a} + \int_b^\infty 2\kappa^2\left\|\Sigma\right\|_{\mathrm{op}}^{-1}\left(1+\frac{6}{cu}\right)\left(1+\frac{p}{5cu}\right)\frac{1}{u^{1-a}}\exp\left(-cu\right)\mathrm{d}u\right)
$$

$$
\leq \frac{10^a\kappa^{2a}}{p^a}\left((cb)^a + 2\kappa^2\left\|\Sigma\right\|_{\mathrm{op}}^{-1}\left(1+\frac{6}{cb}\right)\left(1+\frac{p}{5cb}\right)\frac{1}{(cb)^{1-a}}\exp\left(-cb\right)\right).
$$

Once again this is optimized for $cb = \log(p)$. $\qquad\square$

**Remark 25** (Leverage scores). *Out of simplicity, we only present low rank approximation with random subsampling. Yet, we can improve the result by considering subsampling based on leverage scores. If we consider the Gaussian kernel, $Sk_x \in L^2$ can be thought as a function that is a little bump around $x \in \mathcal{X}$. In essence, subsampling based on leverage scores, consists in representing the solution on a subsampled sequence $(k_{X_i})_{i\in I}$ where the $X_i$ are far from one another so that the bump functions $(Sk_{X_i})$ can approximate a maximum of functions. [45] shows that with leverage scores, we can take $p = (\mu\lambda)^\gamma \log(n)$, with $\gamma$ linked with the capacity of the RKHS linked with the kernel $k$.*

If we add all derivations, we have derived the following theorem.

**Theorem 4.** *Under Assumptions 1, 2 and 3,*

$$
\mathbb{E}_{\mathcal{D}_n}\left[\|\hat{g} - g_\rho\|_{L^2}^2\right]
$$

$$
\leq 8c_{\mathcal{Y}}^2\left(1 + 28\frac{\kappa^2 + \lambda\partial\kappa^2}{\lambda\mu n}\right)(1 + \lambda\mu\|C\|_{\mathrm{op}}^{-1})\frac{\kappa^2 + \lambda\partial\kappa^2}{\lambda\mu}\exp\left(-\frac{\lambda\mu n}{10(\kappa^2 + \lambda\partial\kappa^2)}\right)
$$

$$
\cdots + 4\lambda^2\|\mathcal{L}g_\rho\|_{L^2}^2 + 4\lambda\mu c_a^2\|g_\rho\|_{L^2}^2 + 64\sigma_\ell^2(n_\ell\mu\lambda)^{-1} + 57c_{\mathcal{Y}}^2\kappa^2(n_\ell^2\mu\lambda)^{-1}
$$

$$
\cdots + 64(\kappa^2 + \lambda\partial\kappa^2)^2c_a^2\|g_\rho\|_{L^2}^2(\mu\lambda n)^{-1} + 57(\kappa^2 + \lambda\partial\kappa^2)^2c_a^2\|g_\rho\|_{L^2}^2(\mu\lambda n^2)^{-1} \qquad (36)
$$

$$
\cdots + 12c_a^2\|g_\rho\|_{L^2}^2\left(\frac{10\kappa^2\log(p)}{p} + \frac{10^a\kappa^{2a}c_d\lambda\log(p)^a}{p^a}\right)
$$

$$
\cdots \times \left(1 + \frac{2\kappa^2}{\|\Sigma\|_{\mathrm{op}}\log(p)}\left(1 + \frac{6}{\log(p)}\right)\left(\frac{1}{p} + \frac{1}{5\log(p)}\right)\right).
$$

*where $c_{\mathcal{Y}}$ is an upper bound on $Y$, $\kappa^2$ is an upper bound on $x \to k(x,x)$, $\partial\kappa^2 = \sum_{i=1}^d \kappa_i^2$ with $\kappa_i^2$ a bound on $x \to \partial_{1i}\partial_{2i}k_{x_i}$, $c_d$ and $a$ the constants appearing in Assumption 3, $c_a$ a constant such that $\|g\|_{\mathcal{H}} \leq c_a\|g\|_{L^2}$ and $\sigma_\ell^2 \leq c_{\mathcal{Y}}^2\kappa^2$ a variance parameter linked with the variance of $Y(I+\lambda\mathcal{L})^{-1}\delta_X$.*

Theorem 1 is a corollary of this theorem.