# OpenReview forum: "Overcoming the curse of dimensionality with Laplacian regularization in semi-supervised learning"
_NeurIPS.cc/2021/Conference — NeurIPS 2021 Poster_

### Official Review · Reviewer_UNaU · 2021-07-17

**Rating:** 7
**Confidence:** 2

**Summary:**

This paper considers the problem of semi-supervised learning using Laplacian regularization. The authors use principles and methods from RKHS to mitigate issues with the curse of dimensionality.

**Limitations And Societal Impact:**

Yes

**Main Review:**

The paper appears to be original and is tackling an important problem. The experiments seem correct and support the authors' claim.

My main complaint is that the paper is highly technical. While technical papers are welcomed at NeurIPS, I think it could be made a bit more accessible to the general NeurIPS community. As it stands, it seems the paper will be mostly of interest to those who are deeply in this field. Perhaps some of the technical results could be summarized better with details left for the supplementary material and more experiments moved to the main paper.

**Time Spent Reviewing:**

2

---

> ### Author Response · Authors · 2021-08-07
> **Review answer**
>
> Thank you very much for the time you took to review this paper. We understand your concern about readability. We are convinced that conveying an idea properly is arguably as important as the idea itself, and we will do our best to improve the flow of the paper, and to highlight the intuitions behind the technical mathematical details.

---

### Official Review · Reviewer_YQhQ · 2021-07-17

**Rating:** 8
**Confidence:** 4

**Summary:**

The paper proposes a novel Laplacian regularization method for semi-supervised learning that incorporates an additional penalty term that constraints the learned function to be in a smaller, smoother, RKHS. They characterize their approach via a spectral filtering perspective, provided an approximation that improves the computational complexity in the generalized eigendecomposition, and provided a convergence rate bound under several assumptions that they clearly identified.

**Limitations And Societal Impact:**

Since this is mainly a math/theoretical paper, I don't believe any potential negative societal impact is applicable here.

**Main Review:**

I believe that semi-supervised learning and Laplacian regularization are significant problems of broad interest.

The exposition is clear, and the paper is well organized and well motivated. The technical/mathematical parts of the paper are well written and a pleasure to read.

The authors provided ample background information and motivated their problem clearly, proposed their regularization method, provided different characterizations of their method, and provided a computational approximation that offers a big speedup in computing the eigendecomposition. They outlined their assumptions clearly and provided statistical guarantees.

This is a good quality theory/methods paper.

Several comments:
1. In figure 3b (the subfigure on the right), the computational complexity of small kernel and graph based Laplacian appear to be similar (with the graph-based laplacian appear to be slightly faster than the small kernel)....perhaps in that figure the dimension is not high enough? I would expect that the proposed method with the approximation proposed should scale better computationally than naive Laplacian regularization as dimension scales higher. If so perhaps this can be illustrated with a plot of computational complexity comparison of said methods as you try different dimensions along the x axis?

2. Optimality of bounds: I understand that lower bounds are difficult to show (I certainly don't expect a lower bound to be included in the revision of this paper), but I wonder whether the authors have anything to say about the optimality of their statistical guarantees. Perhaps an easier way to give evidence of how tight their upper bound is is to do some simple simulations under very ideal settings and plot the empirical results along with your bound to show that your bounds are perhaps not (too) loose.

3. Simulations: I understand that this is largely a theory/methods paper, but I do think more simulations/experiments could help make the case more compelling. For one, you mentioned the limitation of the work of El Alaoui et al in part of your motivation but did not provide comparisons against their methods in simulations/experiments.

* A side note is that there is a typo in your reference : Your reference [1] contains only El Alaoui, but the actual paper has many more authors.


**Time Spent Reviewing:**

2 hours

---

> ### Author Response · Authors · 2021-08-07
> **Review answer**
>
> Thank you for your appreciation of this work. We are happy to read that you enjoyed it, and we are really thankful for your comments that have helped us to deepen our understanding of our own work.
>
> **Experiments.** We totally agree that the experimental section can be made thicker, we will do our best to incorporate more experiments to give a better sense of the practical usefulness of our method. Regarding the work of El Alaoui *et al.*, in our understanding, it is useful for regression problems, where the spiky nature of $g_{\text{(naive)}}$ is a problem as illustrated on Figure 4b in Appendix. Yet, when we take its sign it will not be much of a problem. Moreover, the method of El Alaoui implies solving an optimization problem which is prohibitive in terms of computation time.
>
> **Lower bounds.** There are many things to say about bounds.
>
> *1. Defining assumptions for minimax rates.* At high level, lower bounds given by minimax rates are supposed to give an optimal baseline for learning rates. In practice, those bounds depend on assumptions, and are derived by considering the most degenerate function respecting those assumptions. Sometimes a simple well-thought additional assumption can lead to much better bounds. As such, a natural question is what are the natural assumptions to discuss on learnability in a semi-supervised setting. We believe that the source condition (Assumption 1) and its variant based on eigen decomposition of the Laplacian operator is a natural setting to define minimax rates.
>
> *2. Given assumptions, could we have derived better rates?* In terms of optimality of the derivations yielding our bounds, our bounds are partially based on the work of Caponnetto and De Vito, which proves minimax rates for supervised learning, and on the work of Rudi et al. for low-rank approximation for which we are not aware of minimax rates. Hence, we should be able to prove minimax rates without the low-rank approximation. At the end of the day, those bounds boil down to Bernstein concentration inequalities, and optimality of our bounds is highly correlated with optimality of those concentration bounds.
>
> *3. Adaptiveness of our methods to a zoo of assumptions.* Notice that we could prove many more bounds depending on the regularity of the solution and the entropy numbers of the introduced operators. This is due to the fact that kernel methods, contrarily to local averaging methods, are adaptive to regularity of the function to learn.
>
> *4. Limitation of hiding constants.* Finally, it should be highlighted that learning rates are looked at through the exponent raising the number of samples $n$ and not the constant in front of it. Indeed, rates are said to be optimal when they match minimax rates in terms of exponent raising $n$, regardless of constants. Yet, when the number of samples $n$ is fixed, the constants in bounds (typically $\|g\|_{\cal H}$) might be more crucial than the exponent raising $n$.
>
> **Complexity.** Regarding complexity, our implementation of graph-based Laplacian scales in $O(dn^2 + n n_l^2)$. As our method runs in $O(ndp^2)$, you are totally right that taking $p = 50 = \sqrt{2500}$ and $n < 10^3$ is not allowing to show on Figure 3b that our methods can scale faster than graph-based Laplacian. We could show that we can beat graph-Laplacian in terms of speed as $p << \sqrt{n}$. Long-story short, with Figure 3, we wanted to show that our method does not imply extra-computations while yielding much better generalization results when compared to graph-Laplacian.
>
> **Typos.** Thanks for correcting reference [1], this came from an error on DBLP.

---

### Official Review · Reviewer_Cz7V · 2021-07-17

**Rating:** 6
**Confidence:** 3

**Summary:**

This paper addresses the settings of self-supervised learning, providing a new perspective on Laplacian regularization by restricting the solution to a reproducing kernel Hilbert space. They show this formulation leads to new efficient empirical estimates, and demonstrate their approach out-performs the classical graph Laplacian regularization, both in terms of accuracy and computational complexity.


**Limitations And Societal Impact:**

These were addressed in the appendix and not in the main text.

**Main Review:**

The authors provide an interesting new solution to semi-supervised learning, by casting the classical Laplacian regularization typically used into an RKHS perspective. They provide both convergence rates as well as a practical algorithm for their new approach.
My main concern with this paper is that it is ~40 page journal paper trying to squeeze into a dense 9 page conference paper + appendix. As such, the flow of the paper is hard to follow and useful insights and mathematical background that would be of use to the general reader are buried in the appendix, as are experimental results that are more interesting than those presented in the main text. For example, given the limited page length, is Figure 2 essential to the main text? Figures 4 and 5 in the appendix would be more important to demonstrate the performance of the approach.

My second concern is the motivation. The authors claim the graph-based Laplacian is a technique that does not scale well with the dimension. In the text they clarify this statement in 2 places, once in lines 156-157 regarding the scaling of the convergence of the estimator of the graph Laplcian and once in the experimental results in setting the bandwidth of the Gaussian kernel. In practice, regularization with the graph Laplacian has been demonstrated successfully in a broad range of applications with high dimensional data: matrix factorization, convex clustering, dictionary learning, recommender system, data imputation, dimensionality reduction, etc., such that this theoretical concern seems of limited practical implications.

In this context, how dependent are the experimental results on the kernel bandwidth used for the graph Laplacian? In practice, the bandwidth of the Gaussian kernel is typically set in a data-driven manner, such as the median distance of k-nearest neighbors of all points, or using an adaptive bandwidth such as self-tuning bandwidth (Zelnik-Manor and Perona), or multiple other possibilities. The bandwidth used by the author does not seem to depend in any way on the scale of the data itself.

The experiments themselves are synthetic and limited, mostly proof of concept, and more extensive experiments would be required to adopt the author’s approach which the authors state is their goal (I understand this is beyond the scope of the paper).

The paper doesn’t mention recent literature on RKHS in the graph signal processing field for data reconstruction, LMS solutions and spectral filtering:
* SEMI-PARAMETRIC GRAPH KERNEL-BASED RECONSTRUCTION† Vassilis N. Ioannidis, Athanasios N. Nikolakopoulos, and Georgios B. Giannakis
* Kernel-based Reconstruction of Graph Signals Daniel Romero, Meng Ma, Georgios B. Giannakis
* Adaptive Graph Filters in Reproducing Kernel Hilbert Spaces: Design and Performance Analysis, V. R. M. Elias, V. C. Gogineni, W. A. Martins and S. Werner
Additional missing references:
* Semi-supervised learning on Riemannian manifolds, M Belkin, P Niyogi -
* Regularization and semi-supervised learning on large graphs, M Belkin, I Matveeva, P Niyogi




**Time Spent Reviewing:**

3

---

> ### Author Response · Authors · 2021-08-07
> **Review answer**
>
> Thank you very much for the time you took to read and comment on this paper. We are very grateful for your work. It will help us a lot in clarifying, and improving the paper!
>
> **Readability.** We take your concern about readability of the paper really seriously. We will do our best to improve the flow of the paper, and to highlight the intuitions behind the technical mathematical details.
>
> **Comparison with graph Laplacian.** You are completely true, experiments are rather proofs of concept than definitive arguments to drop graph-Laplacian for our method. As such, we did not want to spend too much time on synthetic experiments, while, as you say, graph-Laplacian methods might reserve good surprises on real world applications (even though we are not aware of any results as impressive as self-supervised learning results or supervised learning results). We reserve this detailed experimental comparison to a future and exciting work. This is why we put Figure 2 in the main paper rather than Figures 4 and 5 (that we see as a complement to Figure 3). In our view, Figure 2 is useful to illustrate that the generalized eigenvectors of the matrices ($\Sigma$, $L$) do correspond to the first eigenvectors of the Laplacian.
>
> **Motivations.** In terms of motivations, our idea is to introduce RKHS methods for Laplacian regularization. We strongly believe that reproducing kernel methods are better than local averaging methods, especially because they can circumvent the curse of dimensionality in some cases. We agree that local averaging methods can be applied with success to many problems, but believe that, even for those successes, kernel methods can yield even better results.
>
> **Hyperparameter tuning.** Regarding the bandwidth $\sigma$, we actually chose it to scale with the data, this appear explicitly on Figure 1, but does not appear explicitly on Figure 3 because we chose unit variance Gaussians. Based on our (unreported) experimentations, our method seems much more robust to the choice of the bandwidth than graph-based methods. But as you pointed out, there might be good “self-tuning” methods to improve graph-based methods (note that, similarly, “self-tunning” methods could be designed for reproducing kernels). We will incorporate more of those experiments in the final paper.
>
> **References.** Thank you for the references! We were first planning to refer to literature on graph kernels, and to refer to more papers of Belkin and Niyogi. We finally decided to reduce the number of references to direct the reader to the most relevant ones, but we will happily incorporate them in the final version of the paper.

---

### Decision · Program_Chairs · 2021-09-27

**Decision:**

Accept (Poster)

**Comment:**

The paper received positive reviews. However, only 3 out of 4 reviewers submitted a review, and therefore the AC very carefully read the paper on their own. They do share the positive impressions of the reviewers regarding novelty and they also think that for somebody familiar with the topic this paper is very accessible, while for others it is certainly a hard read. Their major concern regarding the paper is Assumption 1, which unfortunately has not been seen as an issue during the review/rebuttal phase. The main problem they see with Assumption 1 is that it is very restrictive, in fact much more restrictive than a usual source condition, as it explicitly assumes that the target function can be expressed by a finite number of eigenfunctions of the Laplacian, whereas "normal" source conditions are usually phrased in terms of an essentially polynomial decay of the Fourier coefficients. Even worse, the main result, Theorem 1, does not specify the dependence on the number of required eigenfunctions, and a quick look at the appendix did not resolve this issue. As a consequence, the AC views the main result, Theorem 1, as rather premature.

These concerns were discussed with the reviewers and SAC, and it was concluded that this paper should nonetheless be accepted. 1) the authors are attacking a problem of interest to the community; 2) they have some new non-trivial results; 3) the writing is good. The only concern is that the assumption made is not as nice as one might like.